# AN EFFICIENT VARIATIONAL METHOD FOR FITTING LOG-GAUSSIAN COX PROCESSES

## ABSTRACT

Log-Gaussian Cox Processes (LGCP) have been widely used for modeling spatial point patterns. However, fitting LGCP is computationally challenging due to a nested structure involving Poisson process and latent Gaussian random field. To address these issues, we first approximate the intractable LGCP likelihood based on the Voronoi tessellation method. Then, using variational Gaussian approximation, we transform the problem of fitting LGCP into maximizing the evidence lower bound which admits an explicit expression. We design a novel coordinate ascent maximization algorithm which updates the parameter blocks by Newton method and fixed-point method, respectively. To further enhance the computational efficiency, we adopt a nearest neighbor Gaussian process as the prior for the latent Gaussian random field, and the cost of inverting large covariance matrices is greatly reduced via the Woodbury formula. Theoretically, we prove the existence and uniqueness of the optimal solution to the strongly concave objective function, and the convergence of the proposed algorithm is established. Numerical results demonstrate the computational and inferential benefits of our method in modeling log-intensity surface over competing methods.

## 1 INTRODUCTION

Consider a spatial point pattern $\mathbf{Y}$ consisting of a set of observations at locations $\mathcal{D} = \{\mathbf{s}_i\}_{i=1}^N$ in a bounded region $\Omega \subset \mathbb{R}^2$. Log-Gaussian Cox processes (LGCP, Møller et al. (1998)) is a popular point process for modeling the point pattern. Our goal is to infer the LGCP intensity surface $\lambda(\mathbf{s})$ at any location $\mathbf{s}$ given the observed point pattern $\mathbf{Y}$, where $\lambda(\mathbf{s})$ denotes the expected number of points per unit area at $\mathbf{s}$. In LGCP model, the log-intensity surface consists of a linear predictor and a latent Gaussian random field,

$$\log \lambda(\mathbf{s}) = \mathbf{X}(\mathbf{s})\boldsymbol{\beta} + Z(\mathbf{s}), \quad Z(\mathbf{s}) \sim \mathcal{GP}\left(\mathbf{0}, K_{\boldsymbol{\theta}}(\mathbf{s}, \mathbf{s}')\right), \tag{1}$$

where $\mathbf{X}(\mathbf{s})$ is a spatial covariate, $\boldsymbol{\beta}$ is the spatial regression coefficient, and $K_{\boldsymbol{\theta}}$ is the covariance kernel for the Gaussian random field $Z(\mathbf{s})$. Conditional on a realization of the intensity surface $\lambda(\mathbf{s})$, the point pattern $\mathbf{Y}$ follow an inhomogeneous Poisson process. This doubly stochastic, hierarchical construction provides the LGCP with greater flexibility than standard inhomogeneous Poisson models (Illian et al., 2008; Dovers et al., 2023).

The likelihood of LGCP with intensity values $\boldsymbol{\lambda}_{\mathcal{D}} = [\lambda(\mathbf{s}_1), \ldots, \lambda(\mathbf{s}_N)]$ at observed sites $\mathcal{D}$ is

$$p\left(\mathbf{Y} \mid \boldsymbol{\lambda}_{\mathcal{D}}\right) = \exp\left\{-\int_\Omega \lambda(\mathbf{s})\,\mathrm{d}\mathbf{s}\right\}\prod_{i=1}^N \lambda(\mathbf{s}_i), \tag{2}$$

The specific derivation details of the likelihood function can be found in Daley & Vere-Jones (2007). Due to the hierarchical stochastic structure and the existence of the integral term in the likelihood, the problem of fitting LGCP is quite challenging (Murray et al., 2006; Simpson et al., 2016).

**Related work** Many methods have been developed for fitting LGCP. Initially, approaches based on Markov Chain Monte Carlo (MCMC) posterior sampling were explored (Møller et al., 1998; Taylor et al., 2013; Shirota & Gelfand, 2017; Teng et al., 2017). However, while such MCMC-based methods offer theoretical guarantees of asymptotic accuracy, in practice, they are burdened by substantial computational cost and challenging convergence assessment (Taylor & Diggle, 2014).

Various fast approximate algorithms have been developed with computational efficiency. Teng et al. (2017) proposed a variational Bayes mean-field method to approximate the target posterior. Dovers et al. (2023) combined variational inference with low-rank matrix representations of fixed-rank kriging (Cressie & Johannesson, 2008; Zammit-Mangion & Cressie, 2021), yielding substantial speed-ups but limited capacity for modeling local structures (Stein, 2008). Beside the above variational methods, Integrated Nested Laplace Approximation (INLA) introduced by Rue et al. (2009) has become another leading approach to fitting LGCP (Lindgren & Rue, 2015; Simpson et al., 2016; Rue et al., 2017; Bachl et al., 2019; Fuglstad et al., 2019; Flagg & Hoegh, 2023). In particular, Lindgren et al. (2011) enhanced INLA's efficiency by representing the latent Gaussian field as a Gaussian Markov random field. While INLA excels in both accuracy and speed, it focuses on marginal inference and does not directly yield joint posterior distributions (Taylor & Diggle, 2014).

In the machine learning literature, many works have focused on efficiently fitting Gaussian processes. Sparse Gaussian processes and their variants are well studied, such as inducing points (Quinonero-Candela & Rasmussen, 2005; Banerjee et al., 2008) and nearest neighbor GP (NNGP) (Datta et al., 2016). Variational frameworks for these sparse models were subsequently developed by (Titsias, 2009; Hensman et al., 2013; van der Wilk et al., 2020; Wu et al., 2022). Leveraging these advancements, Lloyd et al. (2015) and Shirota & Banerjee (2019) addressed Cox processes where the intensity surface is connected with Gaussian random field via squared and probit link functions, respectively. However, none of the existing works apply to the popular LGCP model which uses the exponential link function.

**Our approach** In this paper, we formulate the problem of fitting LGCP within the Bayesian framework. Based on the observed point pattern, we aim to model the posterior $\lambda(\mathbf{s}) \mid \mathbf{Y}$ of the current log-intensity at any $\mathbf{s}$.

We take a variational approach to modeling the target posterior. First, we approximate the integral term in the LGCP likelihood using a Voronoi tessellation approach. This method not only replaces the original intractable likelihood with an explicit surrogate likelihood, but also incorporates the points of interest as integration nodes. Next, we use variational Gaussian approximation (Challis & Barber, 2013; Wainwright et al., 2008) to obtain the optimal Gaussian approximation to the target posterior by maximizing the evidence lower bound (ELBO). The original problem of posterior distribution inference is thus transformed into an optimization problem with an explicit objective function. Moreover, we prove that the ELBO is strongly concave, and establish the existence and uniqueness of its maximizer.

To solve this optimization problem, we design a novel coordinate ascent maximization algorithm. At each iteration, we update the variational Gaussian mean and the spatial regression coefficient by the Newton method, and update the variational Gaussian covariance matrix by the fixed-point method. To achieve computational efficiency, we approximate the Gaussian random field prior with a nearest-neighbor Gaussian process (NNGP, (Datta et al., 2016)) which induces sparse precision matrix. Additionally, we effectively apply the Woodbury formula to reduce the cost of inverting large matrices. Finally, we establish the convergence of the proposed algorithm with fixed or varying marginal variance hyperparameter and verify the results in numerical experiments.

The rest of the paper is organized as follows. Section 2 introduces the variational method of fitting LGCP, details the Voronoi tessellation based likelihood approximation, constructs the closed-form ELBO, and examines its theoretical properties. Section 3 presents our VoGCAM algorithm for solving the optimization problem and establishes its convergence. Section 4 reports numerical experiments that validate our algorithm's convergence and benchmark our method against existing approaches. All the technical proofs and experimental settings are collected in the Appendix.

## 2 VARIATIONAL INFERENCE FOR FITTING LGCP

### 2.1 LIKELIHOOD APPROXIMATION: A VORONOI TESSELLATION APPROACH

First, we approximate the integral term in LGCP likelihood by numerical integration

$$\int_{\Omega} \lambda(\mathbf{s}) \, d\mathbf{s} \approx \sum_{i=1}^{n} w_i \exp\left\{ \mathbf{X}(\widetilde{\mathbf{s}}_i)\boldsymbol{\beta} + Z(\widetilde{\mathbf{s}}_i) \right\}, \tag{3}$$

where $\mathcal{I} = \{\widetilde{\mathbf{s}}_i\}_{i=1}^n$ represents $n$ pre-selected deterministic integration points in $\Omega$, and $\{w_i\}_{i=1}^n$ are the corresponding weights, satisfying $\sum_{i=1}^n w_i = |\Omega|$. Plugging (3) into the likelihood (2) leads to

$$p(\mathbf{Y} \mid \boldsymbol{\lambda}_{\mathcal{D}}, \boldsymbol{\lambda}_{\mathcal{I}}) \approx \exp\left\{-\sum_{i=1}^n w_i \exp\left(\mathbf{X}(\widetilde{\mathbf{s}}_i)\boldsymbol{\beta} + Z(\widetilde{\mathbf{s}}_i)\right) + \sum_{i=1}^N \left(\mathbf{X}(\mathbf{s}_i)\boldsymbol{\beta} + Z(\mathbf{s}_i)\right)\right\}$$

$$= \exp\left\{-\mathbf{w}^\top \exp\left(\widetilde{\mathbf{X}}\boldsymbol{\beta} + \widetilde{\mathbf{A}}\mathbf{Z}\right) + \mathbf{1}_N^\top\left(\mathbf{X}\boldsymbol{\beta} + \mathbf{A}\mathbf{Z}\right)\right\}, \tag{4}$$

where the matrices $\widetilde{\mathbf{X}}_{n \times m}$ and $\mathbf{X}_{N \times m}$ contain the covariate values at the integration points and the observed points, respectively, with their elements defined as $\widetilde{\mathbf{X}}_{ij} = \mathbf{X}_j(\widetilde{\mathbf{s}}_i)$ and $\mathbf{X}_{ij} = \mathbf{X}_j(\mathbf{s}_i)$. The $(n+N)$-dimensional vector $\mathbf{Z}$ is the realization of $Z(\mathbf{s})$ on $\mathcal{I} \cup \mathcal{D}$, denoted as $\mathbf{Z} = [Z(\widetilde{\mathbf{s}}_1), \dots, Z(\widetilde{\mathbf{s}}_n), Z(\mathbf{s}_1), \dots, Z(\mathbf{s}_N)]^\top$. The selection matrices $\widetilde{\mathbf{A}}_{n \times (n+N)}$ and $\mathbf{A}_{N \times (n+N)}$ are defined as $\widetilde{\mathbf{A}} = [\mathbf{I}_n, \mathbf{0}]$ and $\mathbf{A} = [\mathbf{0}, \mathbf{I}_N]$, respectively; $\mathbf{1}_N$ is a vector of ones of length $N$.

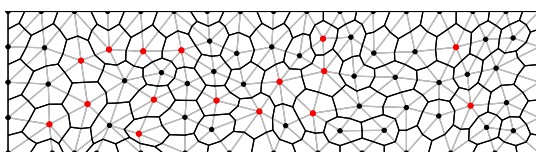

Figure 1: Voronoi tessellation on the observation region. The red solid points denote the target locations $\mathcal{S}$ and the complete set of solid points constitutes the integration locations $\mathcal{I}$ for numerical quadrature. Each integration weight is given by the area of the Voronoi cell around its corresponding integration point, and the gray lines mark the edges of the associated Delaunay triangulation.

In this study, we adopt a Voronoi-tessellation approach to determine the integration points and their corresponding weights (Simpson et al., 2016). First, we perform Delaunay triangulation on the observation area using the positions in the target location set $\mathcal{S}$ as the initial triangle vertices. Next, based on the constructed Delaunay triangulation, we can obtain the Voronoi tessellation that is geometrically dual to it. Finally, we can obtain the integration point set $\mathcal{I}$ and the corresponding weight set $\{w_i\}_{i=1}^n$ as described in Figure 1.

Compared with conventional methods, such as dividing the entire observation domain into a series of uniform grid cells (Teng et al., 2017; Dovers et al., 2023), the Voronoi tessellation offers several advantages. First, it allows for flexible control over the density of the partitioned domain by adjusting the minimum edge length of the Delaunay triangulation. Second, it ensures that information from $\mathcal{S} \setminus \mathcal{D}$ is incorporated into the likelihood function. Third, this approximation method enjoys theoretical guarantee. Simpson et al. (2016) demonstrated that this LGCP likelihood approximation converges to the true likelihood, and further showed that the resulting posterior distribution converges to the true posterior under the Hellinger distance metric. Therefore, substituting the approximated likelihood for the original is reasonable for our subsequent analysis.

## 2.2 The Closed-form ELBO and Theoretical Properties

We begin by treating $\boldsymbol{\beta}$ as a fixed but unknown parameter and then focus on inferring the posterior the latent field realization $\mathbf{Z}_{\mathcal{S}} = [Z(\mathbf{s}_1), \dots, Z(\mathbf{s}_p)]^\top$ at the target sites $\mathcal{S}$. Rather than considering $\mathbf{Z}_{\mathcal{S}}$ only, we infer the full latent vector $\mathbf{Z}$ which combines the values at the integration points $\mathcal{I}$, the observed points $\mathcal{D}$, and the target locations $\mathcal{S}$. The posterior $p(\mathbf{Z} \mid \mathbf{Y})$ offers comprehensive knowledge over the domain besides $\mathbf{Z}_{\mathcal{S}}$. Under the Gaussian field prior $p(\mathbf{Z}) = N(\mathbf{Z} \mid \mathbf{0}, \mathbf{K}_{\boldsymbol{\theta}})$, all the variability in the log-intensity comes from $Z(\mathbf{s})$, and the approximated likelihood (4) can be denoted as $p(\mathbf{Y} \mid \mathbf{Z})$.

Next, we approximate the true posterior $p(\mathbf{Z} \mid \mathbf{Y})$ by a distribution $q^\star$ chosen from a tractable variational family $\mathcal{Q}$ to minimize the Kullback–Leibler divergence $\mathrm{KL}(q(\mathbf{Z}) \| p(\mathbf{Z} \mid \mathbf{Y}))$. The standard variational inference shows that minimizing the KL divergence is equivalent to maximizing ELBO

$$q^\star = \arg\max_{q \in \mathcal{Q}} \int q(\mathbf{Z}) \log \frac{p(\mathbf{Z})p(\mathbf{Y} \mid \mathbf{Z})}{q(\mathbf{Z})} \, d\mathbf{Z}. \tag{5}$$

We choose $\mathcal{Q}$ to be Gaussian such that $q(\mathbf{Z}) = N(\mathbf{Z} \mid \boldsymbol{\mu}, \boldsymbol{\Sigma})$ with the variational parameters $\boldsymbol{\mu}$ and $\boldsymbol{\Sigma}$. Substituting the Gaussian prior $p(\mathbf{Z}) = N(\mathbf{Z} \mid \mathbf{0}, \mathbf{K}_{\boldsymbol{\theta}})$ and the approximated LGCP likelihood $p(\mathbf{Y} \mid \mathbf{Z})$ into the ELBO (5), we obtain an objective function $E(\boldsymbol{\beta}, \boldsymbol{\mu}, \boldsymbol{\Sigma})$ with an explicit form in terms of the unknown parameters $\boldsymbol{\beta}$, $\boldsymbol{\mu}$, and $\boldsymbol{\Sigma}$,

$$
\begin{aligned}
E(\boldsymbol{\beta}, \boldsymbol{\mu}, \boldsymbol{\Sigma}) = & -\mathbf{w}^{\top} \exp\left\{\widetilde{\mathbf{X}}\boldsymbol{\beta} + \widetilde{\mathbf{A}}\boldsymbol{\mu} + \frac{1}{2}\mathrm{diag}\left(\widetilde{\mathbf{A}}\boldsymbol{\Sigma}\widetilde{\mathbf{A}}^{\top}\right)\right\} + \mathbf{1}_N^{\top}(\mathbf{X}\boldsymbol{\beta} + \mathbf{A}\boldsymbol{\mu}) \\
& - \frac{1}{2}\boldsymbol{\mu}^{\top}\mathbf{K}_{\boldsymbol{\theta}}^{-1}\boldsymbol{\mu} \\
& - \frac{1}{2}\mathrm{tr}\left(\mathbf{K}_{\boldsymbol{\theta}}^{-1}\boldsymbol{\Sigma}\right) + \frac{1}{2}\log|\boldsymbol{\Sigma}| - \frac{1}{2}\log|\mathbf{K}_{\boldsymbol{\theta}}| + \frac{n+N}{2},
\end{aligned}
\tag{6}
$$

where $\mathrm{diag}(\cdot)$ is an operator that extracts the diagonal elements of a matrix to form a vector. The derivation of $E(\boldsymbol{\beta}, \boldsymbol{\mu}, \boldsymbol{\Sigma})$ is provided in Appendix A.

We provide an interpretation from the frequentist perspective for $E(\boldsymbol{\beta}, \boldsymbol{\mu}, \boldsymbol{\Sigma})$. The first line in (6) is a goodness-of-fit term, resembling the approximated LGCP log-likelihood (4) only differing by $\frac{1}{2}\mathrm{diag}(\widetilde{\mathbf{A}}\boldsymbol{\Sigma}\widetilde{\mathbf{A}}^{\top})$. The second line penalizes deviation of $\boldsymbol{\mu}$ from $\mathbf{0}$ via a $\mathbf{K}_{\boldsymbol{\theta}}^{-1}$-weighted squared norm, while the third line penalizes $\boldsymbol{\Sigma}$ for deviating from $\mathbf{K}_{\boldsymbol{\theta}}$ via a KL divergence measure; see Lemma 1 for detail. Consequently, maximizing $E(\boldsymbol{\beta}, \boldsymbol{\mu}, \boldsymbol{\Sigma})$ is equivalent to maximizing a penalized likelihood, thereby avoiding overfitting.

The following two results establish the key theoretical properties. Denote by $\mathcal{S}_p^+$ the collection of positive definite matrices of dimension $p$.

**Theorem 1.** *For any prior covariance matrix* $\mathbf{K}_{\boldsymbol{\theta}} \in \mathcal{S}_{n+N}^+$ *, the objective function* $E(\boldsymbol{\beta}, \boldsymbol{\mu}, \boldsymbol{\Sigma})$ *is strictly jointly concave with respect to* $(\boldsymbol{\beta}, \boldsymbol{\mu}, \boldsymbol{\Sigma})$.

The proof of Theorem 1 can be directly obtained from the expression of $E(\boldsymbol{\beta}, \boldsymbol{\mu}, \boldsymbol{\Sigma})$ in (6), combined with the fact that both $-\exp\{\cdot\}$ and $\log|\cdot|$ are operations that preserve strong concavity.

Based on the strict concavity established by Theorem 1, we prove that the maximizer of the objective function $E(\boldsymbol{\beta}, \boldsymbol{\mu}, \boldsymbol{\Sigma})$ uniquely exists. Its proof is in Appendix B.

**Theorem 2.** *The optimization problem*

$$
\max_{\boldsymbol{\beta}\in\mathrm{R}^m, \boldsymbol{\mu}\in\mathrm{R}^{n+N}, \boldsymbol{\Sigma}\in\mathcal{S}_{n+N}^+} E(\boldsymbol{\beta}, \boldsymbol{\mu}, \boldsymbol{\Sigma})
\tag{7}
$$

*admits a unique solution, denoted by* $(\boldsymbol{\beta}^\star, \boldsymbol{\mu}^\star, \boldsymbol{\Sigma}^\star)$.

## 3 EFFICIENT ALGORITHM FOR OPTIMIZING THE ELBO

### 3.1 APPROXIMATE PRIOR BY NNGP

In each iteration of our algorithm, evaluating the ELBO $E(\boldsymbol{\beta}, \boldsymbol{\mu}, \boldsymbol{\Sigma})$ requires operations on the $(n+N)$-dimensional prior covariance $\mathbf{K}_{\boldsymbol{\theta}}$, which incur $\mathcal{O}((n+N)^3)$ time and $\mathcal{O}((n+N)^2)$ storage cost. To alleviate this burden, we approximate the Gaussian process $Z(\mathbf{s})$ by a NNGP, in which each full conditional distributions is replaced by one conditioned on a small set of nearest neighbors. The resulting finite-dimensional prior has a sparse precision matrix, yielding the approximate prior

$$
p(\mathbf{Z}) \approx \widetilde{p}(\mathbf{Z}) = N\left(\mathbf{Z} \mid \mathbf{0}, \sigma^2 \boldsymbol{\Gamma}^{-1}\right).
\tag{8}
$$

where $\sigma^2$ is the marginal variance and $\boldsymbol{\Gamma}$ is a sparse precision matrix parameterized by hyperparameters including range and smoothness. Further details on constructing $\boldsymbol{\Gamma}$ are provided in Appendix C.1.

### 3.2 COORDINATE ASCENT MAXIMIZATION ALGORITHM

Substituting the NNGP approximate covariance matrix $\sigma^2 \boldsymbol{\Gamma}^{-1}$ into the ELBO expression (6) yields the current objective function, which is given by

$$
\begin{aligned}
E_{\boldsymbol{\theta}}(\boldsymbol{\beta}, \boldsymbol{\mu}, \boldsymbol{\Sigma}) = & -\mathbf{w}^{\top} \exp\left\{\widetilde{\mathbf{X}}\boldsymbol{\beta} + \widetilde{\mathbf{A}}\boldsymbol{\mu} + \frac{1}{2}\mathrm{diag}\left(\widetilde{\mathbf{A}}\boldsymbol{\Sigma}\widetilde{\mathbf{A}}^{\top}\right)\right\} + \mathbf{1}_N^{\top}(\mathbf{X}\boldsymbol{\beta} + \mathbf{A}\boldsymbol{\mu}) \\
& - \frac{\sigma^{-2}}{2}\boldsymbol{\mu}^{\top}\boldsymbol{\Gamma}\boldsymbol{\mu} - \frac{\sigma^{-2}}{2}\mathrm{tr}(\boldsymbol{\Gamma}\boldsymbol{\Sigma}) + \frac{1}{2}\log|\boldsymbol{\Sigma}| - \frac{1}{2}\log|\sigma^2\boldsymbol{\Gamma}^{-1}| + \frac{n+N}{2}.
\end{aligned}
\tag{9}
$$

We first consider the case where the hyperparameters $\boldsymbol{\theta}$ defining $\sigma^2$ and $\boldsymbol{\Gamma}$ are fixed. Then, the optimization problem is given by

$$\max_{\boldsymbol{\beta}\in\mathrm{R}^m,\boldsymbol{\mu}\in\mathrm{R}^{n+N},\boldsymbol{\Sigma}\in\mathcal{S}_{n+N}^+} E_{\boldsymbol{\theta}}\left(\boldsymbol{\beta},\boldsymbol{\mu},\boldsymbol{\Sigma}\right). \tag{10}$$

Since the parameters $\boldsymbol{\beta}$, $\boldsymbol{\mu}$, and $\boldsymbol{\Sigma}$ are coupled in the objective function, we employ block coordinate ascent method: at each iteration, we update $\boldsymbol{\beta}$ (with $\boldsymbol{\mu}$, $\boldsymbol{\Sigma}$ fixed) and $\boldsymbol{\mu}$ (with $\boldsymbol{\beta}$, $\boldsymbol{\Sigma}$ fixed) via Newton method, then update $\boldsymbol{\Sigma}$ (with $\boldsymbol{\beta}$, $\boldsymbol{\mu}$ fixed) via a fixed-point method.

**Newton method for updating $\boldsymbol{\beta}$ and $\boldsymbol{\mu}$.** The gradients of the objective function $E_{\boldsymbol{\theta}}\left(\boldsymbol{\beta},\boldsymbol{\mu},\boldsymbol{\Sigma}\right)$ with respect to $\boldsymbol{\beta}$ and $\boldsymbol{\mu}$ are given by

$$\begin{aligned}
\nabla_{\boldsymbol{\beta}}E &= \mathbf{X}^\top\mathbf{1}_N - \widetilde{\mathbf{X}}^\top\left[\mathbf{w}\circ\exp\left\{\widetilde{\mathbf{X}}\boldsymbol{\beta}+\widetilde{\mathbf{A}}\boldsymbol{\mu}+\frac{1}{2}\mathrm{diag}\left(\widetilde{\mathbf{A}}\boldsymbol{\Sigma}\widetilde{\mathbf{A}}^\top\right)\right\}\right], \\
\nabla_{\boldsymbol{\mu}}E &= \mathbf{A}^\top\mathbf{1}_N - \widetilde{\mathbf{A}}^\top\left[\mathbf{w}\circ\exp\left\{\widetilde{\mathbf{X}}\boldsymbol{\beta}+\widetilde{\mathbf{A}}\boldsymbol{\mu}+\frac{1}{2}\mathrm{diag}\left(\widetilde{\mathbf{A}}\boldsymbol{\Sigma}\widetilde{\mathbf{A}}^\top\right)\right\}\right] - \sigma^{-2}\boldsymbol{\Gamma}\boldsymbol{\mu}.
\end{aligned} \tag{11}$$

The Hessian matrices of $E_{\boldsymbol{\theta}}\left(\boldsymbol{\beta},\boldsymbol{\mu},\boldsymbol{\Sigma}\right)$ with respect to $\boldsymbol{\beta}$ and $\boldsymbol{\mu}$ are given by

$$\begin{aligned}
\nabla_{\boldsymbol{\beta}}^2 E &= -\widetilde{\mathbf{X}}^\top\mathrm{Diag}\left[\mathbf{w}\circ\exp\left\{\widetilde{\mathbf{X}}\boldsymbol{\beta}+\widetilde{\mathbf{A}}\boldsymbol{\mu}+\frac{1}{2}\mathrm{diag}\left(\widetilde{\mathbf{A}}\boldsymbol{\Sigma}\widetilde{\mathbf{A}}^\top\right)\right\}\right]\widetilde{\mathbf{X}}, \\
\nabla_{\boldsymbol{\mu}}^2 E &= -\widetilde{\mathbf{A}}^\top\mathrm{Diag}\left[\mathbf{w}\circ\exp\left\{\widetilde{\mathbf{X}}\boldsymbol{\beta}+\widetilde{\mathbf{A}}\boldsymbol{\mu}+\frac{1}{2}\mathrm{diag}\left(\widetilde{\mathbf{A}}\boldsymbol{\Sigma}\widetilde{\mathbf{A}}^\top\right)\right\}\right]\widetilde{\mathbf{A}} - \sigma^{-2}\boldsymbol{\Gamma}.
\end{aligned} \tag{12}$$

Here, $\mathbf{x}\circ\mathbf{y}$ denotes the Hadamard product of vectors $\mathbf{x}$ and $\mathbf{y}$. The operator $\mathrm{Diag}(\cdot)$ acts on a vector, constructing a diagonal matrix with the elements of that vector on its main diagonal.

Given initial values $\boldsymbol{\beta}_0$ and $\boldsymbol{\mu}_0$, we update $\boldsymbol{\beta}$ and $\boldsymbol{\mu}$ using Newton method. The updating rules at the $k$-th iteration are given by

$$\boldsymbol{\beta}_{k+1} = \boldsymbol{\beta}_k - (\nabla_{\boldsymbol{\beta}}^2 E_k)^{-1}\nabla_{\boldsymbol{\beta}}E_k, \quad \boldsymbol{\mu}_{k+1} = \boldsymbol{\mu}_k - (\nabla_{\boldsymbol{\mu}}^2 E_k)^{-1}\nabla_{\boldsymbol{\mu}}E_k. \tag{13}$$

The major computational cost of Newton method lies in solving the linear systems involving the Hessians. Since the negative Hessians are positive definite, these systems can be solved efficiently using the conjugate gradient method in practice.

**Fixed-point method for updating $\boldsymbol{\Sigma}$.** For the matrix variable $\boldsymbol{\Sigma}$, using Newton method for updates would involve solving a large-scale linear system. Instead, we design a fixed-point iteration method. Since the objective function $E_{\boldsymbol{\theta}}\left(\boldsymbol{\beta},\boldsymbol{\mu},\boldsymbol{\Sigma}\right)$ is strongly concave with respect to $\boldsymbol{\Sigma}$, finding the optimal $\boldsymbol{\Sigma}^\star$ is equivalent to solving the equation $\nabla_{\boldsymbol{\Sigma}}E = \mathbf{0}$:

$$-\widetilde{\mathbf{A}}^\top\mathrm{Diag}\left[\mathbf{w}\circ\exp\left\{\widetilde{\mathbf{X}}\boldsymbol{\beta}+\widetilde{\mathbf{A}}\boldsymbol{\mu}+\frac{1}{2}\mathrm{diag}\left(\widetilde{\mathbf{A}}\boldsymbol{\Sigma}\widetilde{\mathbf{A}}^\top\right)\right\}\right]\widetilde{\mathbf{A}} - \sigma^{-2}\boldsymbol{\Gamma} + \boldsymbol{\Sigma}^{-1} = \mathbf{0}. \tag{14}$$

See Lemma 2 in Appendix C.2 for its derivation.

Based on (14), we construct a fixed-point method to update $\boldsymbol{\Sigma}$. Specifically, given an initial value $\boldsymbol{\Sigma}_0$, the updating rule at the $k$-th iteration is given by

$$\boldsymbol{\Sigma}_{k+1} = \left(\mathbf{D}_k + \sigma^{-2}\boldsymbol{\Gamma}\right)^{-1}, \tag{15}$$

where $\mathbf{D}_k = \widetilde{\mathbf{A}}^\top\mathrm{Diag}\left[\mathbf{w}\circ\exp\left\{\widetilde{\mathbf{X}}\boldsymbol{\beta}+\widetilde{\mathbf{A}}\boldsymbol{\mu}+\frac{1}{2}\mathrm{diag}\left(\widetilde{\mathbf{A}}\boldsymbol{\Sigma}_k\widetilde{\mathbf{A}}^\top\right)\right\}\right]\widetilde{\mathbf{A}}$. The sequence $\{\boldsymbol{\Sigma}_k\}_{k\geq 0}$ generated by (15) converges. Its proof is in Theorem 4 in Appendix C.3.

**Matrix inversion with the Woodbury formula.** In each iteration, the dominant cost arises from inverting the $(n+N)$-dimensional matrix $\left(\mathbf{D}_k + \sigma^{-2}\boldsymbol{\Gamma}\right)^{-1}$ in (15), which scales as $\mathcal{O}((n+N)^3)$. To reduce the complexity, we employ the Woodbury formula after decomposing the precision matrix into a diagonal plus low-rank form,

$$\boldsymbol{\Gamma} \approx \boldsymbol{\Gamma}_{\mathrm{diag}} + \mathbf{L}\mathbf{L}^\top, \tag{16}$$

where $\Gamma_{\text{diag}}$ is chosen so that $\Gamma - \Gamma_{\text{diag}}$ remains positive definite, and $\mathbf{L}_{(N+n)\times r}$ (with $r \ll n + N$) derives from its Cholesky factor. It is necessary to retain the diagonal component. By its definition (15), $\mathbf{D}_k$ is an $(n + N)$-dimensional diagonal matrix whose last $N$ diagonal elements are zero. The diagonal $\Gamma_{\text{diag}}$ ensures that $\widetilde{\mathbf{D}}_k$ in (17) is invertible such that the Woodbury formula is applicable.

With the approximation of $\Gamma$ in (16), the fixed-point iteration is

$$\mathbf{\Sigma}_{k+1} \approx \left(\mathbf{D}_k + \sigma^{-2}\Gamma_{\text{diag}} + \sigma^{-2}\mathbf{L}\mathbf{L}^\top\right)^{-1} \triangleq \left(\widetilde{\mathbf{D}}_k + \sigma^{-2}\mathbf{L}\mathbf{L}^\top\right)^{-1}, \tag{17}$$

where $\widetilde{\mathbf{D}}_k = \mathbf{D}_k + \sigma^{-2}\Gamma_{\text{diag}}$ is an invertible diagonal matrix. It follows from the Woodbury formula that

$$\mathbf{\Sigma}_{k+1} = \widetilde{\mathbf{D}}_k^{-1} - \sigma^{-2}\widetilde{\mathbf{D}}_k^{-1}\mathbf{L}\left(\mathbf{I}_r + \sigma^{-2}\mathbf{L}^\top\widetilde{\mathbf{D}}_k^{-1}\mathbf{L}\right)^{-1}\mathbf{L}^\top\widetilde{\mathbf{D}}_k^{-1}. \tag{18}$$

The complexity for each update becomes $\mathcal{O}\left((n + N)^2 r + (n + N)r^2 + r^3\right)$. It is significantly more efficient than the cost of the direct matrix inversion $\mathcal{O}\left((n + N)^3\right)$.

**Algorithm and its convergence guarantee.** When all hyperparameters $\boldsymbol{\theta}$ are fixed, the process of solving the optimization problem (10) is detailed in Algorithm 1.

---

**Algorithm 1:** VoGCAM with fixed hyperparameters

---

**Input:** Covariate matrices $\mathbf{X}$, $\widetilde{\mathbf{X}}$; Selection matrices $\mathbf{A}$, $\widetilde{\mathbf{A}}$;
      Dual mesh weights $\mathbf{w}$; Fixed hyperparameters $\boldsymbol{\theta}$;
      NNGP approximate precision matrix $\Gamma$.
**Approximate $\Gamma$** by (16)
**Initialize** $\widehat{\boldsymbol{\beta}}_0 \leftarrow \boldsymbol{\beta}_0$; $\widehat{\boldsymbol{\mu}}_0 \leftarrow \boldsymbol{\mu}_0$; $\widehat{\mathbf{\Sigma}}_0 \leftarrow \mathbf{\Sigma}_0$
**while** $E_{\boldsymbol{\theta}}\left(\widehat{\boldsymbol{\beta}}_k, \widehat{\boldsymbol{\mu}}_k, \widehat{\mathbf{\Sigma}}_k\right)$ not converged and $k \leq k_{max}$ **do:**
    **Update** $\widehat{\boldsymbol{\beta}}_k$ and $\widehat{\boldsymbol{\mu}}_k$ by Newton method (13)
    **Update** $\widehat{\mathbf{\Sigma}}_k$ by fixed-point method (18)
**return** $\widehat{\boldsymbol{\beta}}^\star, \widehat{\boldsymbol{\mu}}^\star, \widehat{\mathbf{\Sigma}}^\star$

---

The convergence analysis of Algorithm 1 is provided in Appendix C.4.

## 3.3 HYPERPARAMETER CONFIGURATION

Prior to running the algorithm, we first need to fix all hyperparameters defining the precision matrix $\Gamma$ (e.g. the Matérn range and smoothness) using strategies such as K-fold cross-validation (Finley et al., 2019) or empirical estimation methods based on the spatial properties (Lindgren et al., 2011). The practical procedure for determining the aforementioned hyperparameters is detailed in Appendix E. Then, we turn to estimation of the marginal variance $\sigma^2$.

**The new ELBO that contains $\sigma$** To incorporate hierarchical modeling of the marginal variance $\sigma^2$, we augment the ELBO in (5) by including the prior $p(\sigma^2 \mid \boldsymbol{\alpha}_\sigma)$, which yields

$$E\left(\boldsymbol{\beta}, \boldsymbol{\mu}, \mathbf{\Sigma}; \sigma\right) = \int q(\mathbf{Z}) \log \frac{p\left(\mathbf{Z}, \mathbf{Y} \mid \sigma^2\right) p\left(\sigma^2 \mid \boldsymbol{\alpha}_\sigma\right)}{q(\mathbf{Z})}\, \mathrm{d}\mathbf{Z}$$

$$= E_{\boldsymbol{\theta}}\left(\boldsymbol{\beta}, \boldsymbol{\mu}, \mathbf{\Sigma}\right) + \log p\left(\sigma^2 \mid \boldsymbol{\alpha}_\sigma\right), \tag{19}$$

where $\boldsymbol{\alpha}_\sigma$ collects the hyperparameters of the prior on $\sigma^2$. In particular, we place an inverse-Gamma prior, $p\left(\sigma^2 \mid \boldsymbol{\alpha}_\sigma\right) = \text{IG}\left(\sigma^2 \mid a, b\right)$, with shape $a$ and scale $b$. Next, combining $E_{\boldsymbol{\theta}}\left(\boldsymbol{\beta}, \boldsymbol{\mu}, \mathbf{\Sigma}\right)$ in (9) and the inverse-Gamma prior for $\sigma^2$, we write (19) as

$$E\left(\boldsymbol{\beta}, \boldsymbol{\mu}, \mathbf{\Sigma}; \sigma\right) = -\mathbf{w}^\top \exp\left\{\widetilde{\mathbf{X}}\boldsymbol{\beta} + \widetilde{\mathbf{A}}\boldsymbol{\mu} + \frac{1}{2}\text{diag}\left(\widetilde{\mathbf{A}}\mathbf{\Sigma}\widetilde{\mathbf{A}}^\top\right)\right\} + \mathbf{1}_N^\top\left(\mathbf{X}\boldsymbol{\beta} + \mathbf{A}\boldsymbol{\mu}\right)$$

$$- \frac{\sigma^{-2}}{2}\boldsymbol{\mu}^\top\Gamma\boldsymbol{\mu} - \frac{\sigma^{-2}}{2}\text{tr}\left(\Gamma\mathbf{\Sigma}\right) + \frac{1}{2}\log|\mathbf{\Sigma}| - \frac{1}{2}\log|\sigma^2\Gamma^{-1}| + \frac{n + N}{2} \tag{20}$$

$$+ (a + 1)\log\sigma^{-2} + b\sigma^{-2} + b\log a - \log\Gamma(a).$$

---

**Algorithm 2:** VoGCAM

---

**Input:** Covariate matrices $\mathbf{X}, \widetilde{\mathbf{X}}$; Selection matrices $\mathbf{A}, \widetilde{\mathbf{A}}$;
      Dual mesh weights $\mathbf{w}$; Fixed hyperparameters $\boldsymbol{\theta}$ except for $\sigma$;
      NNGP approximate precision matrix $\boldsymbol{\Gamma}$.
**Approximate $\boldsymbol{\Gamma}$ by (16)**
**Initialize** $\widehat{\boldsymbol{\beta}}_0 \leftarrow \boldsymbol{\beta}_0; \widehat{\boldsymbol{\mu}}_0 \leftarrow \boldsymbol{\mu}_0; \widehat{\boldsymbol{\Sigma}}_0 \leftarrow \boldsymbol{\Sigma}_0 ; \widehat{\sigma}_0 \leftarrow \sigma_0$
**while** $E\left(\widehat{\boldsymbol{\beta}}_k, \widehat{\boldsymbol{\mu}}_k, \widehat{\boldsymbol{\Sigma}}_k; \widehat{\sigma}_k\right)$ not converged and $k \leq k_{max}$ **do:**
    **Update** $\widehat{\boldsymbol{\beta}}_k, \widehat{\boldsymbol{\mu}}_k$ and $\widehat{\boldsymbol{\Sigma}}_k$ by algorithm 1
    **Update** $\widehat{\sigma}_k$ by fixed-point method (21)
**return** $\widehat{\boldsymbol{\beta}}^\star, \widehat{\boldsymbol{\mu}}^\star, \widehat{\boldsymbol{\Sigma}}^\star, \widehat{\sigma}^\star$

---

**Fixed-pointed method for updating $\sigma$.** To maximize $E(\boldsymbol{\beta}, \boldsymbol{\mu}, \boldsymbol{\Sigma}; \sigma)$, we partition the parameters into two blocks, $(\boldsymbol{\beta}, \boldsymbol{\mu}, \boldsymbol{\Sigma})$ and $\sigma$, and then alternately update the two blocks. Specifically, in the $k$-th iteration, we first fix $\sigma$ at its current value, $\sigma_k$. Next, as described in Section 3.2, we can find the optimal values that maximize $E_{\boldsymbol{\theta}_k}(\boldsymbol{\beta}, \boldsymbol{\mu}, \boldsymbol{\Sigma})$, denoted as $(\boldsymbol{\beta}_{\sigma_k}^\star, \boldsymbol{\mu}_{\sigma_k}^\star, \boldsymbol{\Sigma}_{\sigma_k}^\star)$. Then, fixing $(\boldsymbol{\beta}, \boldsymbol{\mu}, \boldsymbol{\Sigma})$ at these optimal values, we update $\sigma$ by solving $\partial E\left(\boldsymbol{\beta}_{\sigma_k}^\star, \boldsymbol{\mu}_{\sigma_k}^\star, \boldsymbol{\Sigma}_{\sigma_k}^\star; \sigma\right)/\partial(\sigma^2) = 0$. This yields a fixed-point iteration update

$$\sigma_{k+1} = \sqrt{\frac{\boldsymbol{\mu}_{\sigma_k}^\top \boldsymbol{\Gamma} \boldsymbol{\mu}_{\sigma_k} + \text{tr}\left(\boldsymbol{\Gamma}\boldsymbol{\Sigma}_{\sigma_k}\right) - 2b}{n + N + 2a + 2}}. \tag{21}$$

We have obtained the complete algorithm summarized in Algorithm 2. The following theorem guarantees the convergence of the update for $\sigma$. Its proof is provided in Appendix D.

**Theorem 3.** *For any initial value $\sigma_0 > 0$, the sequence $\{\sigma_k\}_{k \geq 0}$ generated by (21) converges monotonically to some $\sigma_\star > 0$.*

## 4 EXPERIMENTS

Our method is termed as VoGCAM, short for **V**ariational **V**oronoi **G**aussian **C**oordinate **A**scent **M**aximization. In the subsequent simulation and real data, our method will be compared with INLA (Simpson et al., 2016; Fuglstad et al., 2019) (implemented using the `INLA` package) and VIFRK (Dovers et al., 2023) (implemented using the `scampr` package), which are already the two best-performing methods for fitting LGCP in spatial statistics.

**Simulation.** We first generate a set of point patterns simulated by LGCP on the bounded region $\Omega = [0, 10] \times [0, 10]$, as shown in Figure 2 (a). Based on the current point pattern, we consider two types of cases in the following experiments. The first type is the conventional scenario that we can observe all point patterns, as shown in Figure 2 (b), which we denote as the "full case". The second type is the scenario that considers the sampling effort, as shown in Figure 2 (c) (Simpson et al., 2016; Flagg & Hoegh, 2023). We suppose that all points within the red border and some points outside the border cannot be observed, and we denote this as the "hole case". More details on the experiment settings and the convergence verification of the VoGCAM algorithm are presented in Appendix E.

We evaluate VoGCAM against INLA and VIFRK (Dovers et al., 2023) under two scenarios, "full case" and "hole case". To capture posterior information throughout $\Omega$, we take the target set $\mathcal{S}$ to be the integration points $\mathcal{I}$ defined by the Voronoi tessellation. The comparison results are shown in Figure 3.

We then evaluated each method's performance in both the "full case" and "hole case" by computing the log-likelihood at the observed locations set $\mathcal{D}$ and the predictive likelihood at the target locations set $\mathcal{S}$. To assess computational efficiency, we also recorded the runtime required for model fitting. The results are presented in Table 1.

In the left panel of Figure 3, all the three methods recover the LGCP intensity surface accurately in the "full case" according to their posterior mean fits. However, under the more challenging "hole

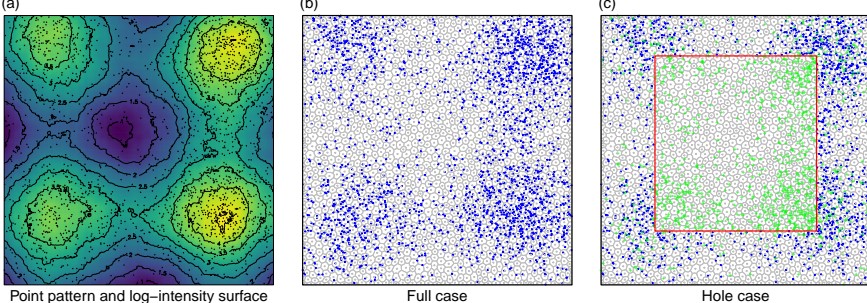

Figure 2: (a) represents a set of point patterns (black solid dots) generated by LGCP simulation on $\Omega$ and its corresponding LGCP log-intensity surface; (b) and (c) represent the distribution of observed points and the Voronoi tessellation on the observation area under the "full case" and "hole case" respectively. The blue solid dots are the observed points, and the green dots represent the points in the original point pattern that were not observed.

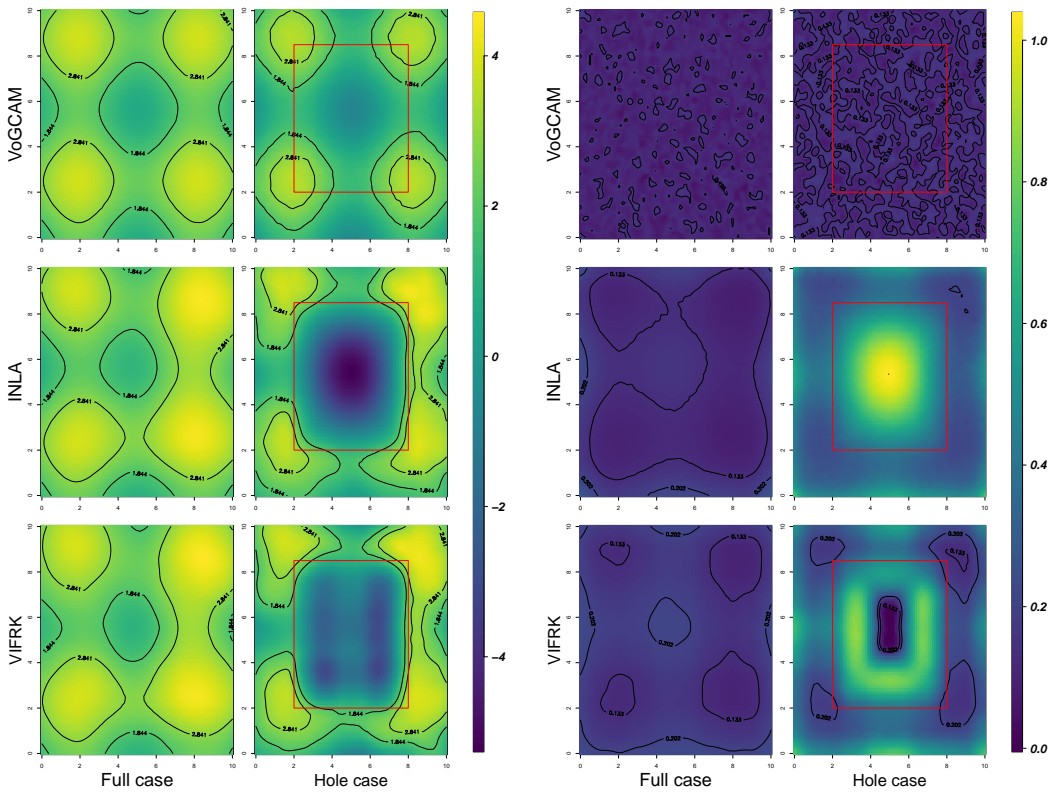

Figure 3: Comparison of the posterior mean surface (left figure) and posterior standard error surface (right figure) of the log-intensity fitted by the three methods based on the observation points in the two scenarios of "full case" and "hole case".

case" where a substantial portion of the observations is missing, only our approach succeeds in tracking the true LGCP intensity closely, while the performance of INLA and VIFRK deteriorates markedly. Comparison of posterior standard errors in the right panel of Figure 3 reconciles the same conclusion via uncertainty quantification.

**Real Data.** We further use two real-world datasets to validate the efficiency of the proposed method. The first neuronal dataset consists of 583 training data and 29127 testing data. It was originally analyzed by Aglietti et al. (2019) where detailed experiment setting can be found. In the

Table 1: Comparison of the observed log-likelihood function (obs loglik) at the observation point $\mathcal{D}$, the predicted log-likelihood function (pred loglik) at the prediction point $\mathcal{S}$, and the seconds of time consumption for three different methods in the two scenarios of "full case" and "hole case".

| Method | Full case | | | Hole case | | |
|---|---|---|---|---|---|---|
| | obs loglik | pred loglik | time | obs loglik | pred loglik | time |
| VoGCAM | 5423.12 | 2505.15 | 6.51 | 2739.40 | 1958.95 | 5.73 |
| INLA | 5533.71 | 2338.11 | 10.32 | 2717.11 | 749.41 | 9.55 |
| VIFRK | 5534.91 | 2361.77 | 0.20 | 2709.50 | 904.28 | 0.11 |

second real-data application, we analyzed spatially resolved transcriptomics data generated by the Xenium platform (10x Genomics), which provides subcellular resolution expression measurements of hundreds of genes directly in tissue sections. We focused on the breast cancer S1R1 sample released by 10x Genomics and cropped a region containing approximately 5,000 cells. In our point process formulation, the response was the expression of the epithelial marker KRT15, while two covariates captured local microenvironmental context: the average abundance of DCIS_1 cells and the average abundance of Myoepi_KRT15$^+$ cells in the neighborhood of each spatial point. Our method estimated the spatially varying log-intensity surface of KRT15 expression and quantified its dependence on these local covariates, to extract interpretable gene–microenvironment associations from high-resolution spatial transcriptomics data.

Table 2: Comparison of the observed log-likelihood (obs loglik), predicted log-likelihood (pred loglik), and computation time for different methods on two real-world datasets.

| Method | Neuronal data | | | Transcriptomics data | | |
|---|---|---|---|---|---|---|
| | obs loglik | pred loglik | time | obs loglik | pred loglik | time |
| VoGCAM | $-3699.77$ | $-38231.83$ | 9.51 | 36404.93 | 16430.17 | 12.17 |
| VIFRK | $-2106.14$ | $-82399.21$ | 2.06 | 43955.61 | 6147.24 | 0.13 |

We applied our proposed VoGCAM and competing methods, including INLA and VIFRK, to the datasets. Table 2 presents the results of observed and predicted likelhoods, and computing time. Since the number of prediction points in real data is much larger than that of training points, INLA encountered severe numerical ill-conditioning problems during training, leading to training failure. When comparing our method with VIFRK, we find that VoGCAM performs better than VIFRK in terms of predicted likelihood while its observed likelihood metric is inferior. As expected, VIFRK excels in computing time due to its focus on global feature. In addition, according to the experimental results shown in Table 3 of Aglietti et al. (2019), the predicted loglikelihood obtained by their proposed STVB method is $-84550$ with a running time of $193.07$ seconds. In summary, since the primary goal in this work is the prediction task for LGCP intensity function at unobserved spatial locations, the proposed VoGCAM is a computationally efficient method for modeling complex spatial point pattern data.

## 5 DISCUSSION

In this paper, we propose the VoGCAM method to efficiently fit the LGCP. By using the variational Gaussian approximation, we transform the challenging LGCP inference problem into an optimization problem. Moreover, we propose a novel and efficient coordinate ascent maximization algorithm to solve this optimization problem with theoretical guarantee. Numerical experiments also validate the effectiveness of our algorithm. Finally, we also discuss some limitations of our method, see Appendix F.

## REPRODUCIBILITY STATEMENT

To ensure the reproducibility of our work, we have provided a comprehensive set of resources. All code necessary to replicate the experiments and generate the results presented in this paper is submitted as supplementary materials. The datasets used are either publicly available or simulated, with clear descriptions of their sources and generating processes. For theoretical results, our appendix contains all necessary proofs and a clear explanation of any assumptions made.

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

## A  DERIVATION OF THE EXPLICIT EXPRESSION OF ELBO

First, we derive the explicit expression of $E(\boldsymbol{\beta}, \boldsymbol{\mu}, \boldsymbol{\Sigma})$ in 6. Based on the definition of ELBO in 5, we have:

$$
\begin{aligned}
E(\boldsymbol{\beta}, \boldsymbol{\mu}, \boldsymbol{\Sigma}) &= \int q(\mathbf{Z}) \log \frac{p(\mathbf{Y} \mid \mathbf{Z}) p(\mathbf{Z})}{q(\mathbf{Z})} \, d\mathbf{Z} \\
&= \int q(\mathbf{Z}) \log p(\mathbf{Y} \mid \mathbf{Z}) \, d\mathbf{Z} + \int q(\mathbf{Z}) \log p(\mathbf{Z}) \, d\mathbf{Z} - \int q(\mathbf{Z}) \log q(\mathbf{Z}) \, d\mathbf{Z} \\
&= \mathrm{E}_{q(\mathbf{Z})} \left[ \log p(\mathbf{Y} \mid \mathbf{Z}) \right] + \mathrm{E}_{q(\mathbf{Z})} \left[ \log p(\mathbf{Z}) \right] - \mathrm{E}_{q(\mathbf{Z})} \left[ \log q(\mathbf{Z}) \right].
\end{aligned} \tag{22}
$$

Next, we will calculate the explicit expressions of the three parts in (22) separately. For the derivation of the first part, we use the conclusion of the moment–generating function in the calculation. The specific details are as follows:

$$
\begin{aligned}
\mathrm{E}_{q(\mathbf{Z})} \left[ \log p(\mathbf{y} \mid \mathbf{Z}) \right] &= \mathrm{E}_{N(\mathbf{Z} \mid \boldsymbol{\mu}, \boldsymbol{\Sigma})} \left[ -\mathbf{w}^\top \exp\left( \widetilde{\mathbf{X}}\boldsymbol{\beta} + \widetilde{\mathbf{A}}\mathbf{Z} \right) + \mathbf{1}_n^\top (\mathbf{X}\boldsymbol{\beta} + \mathbf{A}\mathbf{Z}) \right] \\
&= -\mathbf{w}^\top \exp\left( \widetilde{\mathbf{X}}\boldsymbol{\beta} \right) \mathrm{E}_{N(\mathbf{Z} \mid \boldsymbol{\mu}, \boldsymbol{\Sigma})} \left[ \sum_{i=1}^{n} \exp\left( \widetilde{\mathbf{A}}_i \mathbf{Z} \right) \right] + \mathbf{1}_N^\top (\mathbf{X}\boldsymbol{\beta} + \mathbf{A}\boldsymbol{\mu}) \\
&= -\mathbf{w}^\top \exp\left( \widetilde{\mathbf{X}}\boldsymbol{\beta} \right) \cdot \sum_{i=1}^{n} \exp\left( \widetilde{\mathbf{A}}_i^\top \boldsymbol{\mu} + \frac{1}{2} \widetilde{\mathbf{A}}_i \boldsymbol{\Sigma} \widetilde{\mathbf{A}}_i^\top \right) + \mathbf{1}_N^\top (\mathbf{X}\boldsymbol{\beta} + \mathbf{A}\boldsymbol{\mu}) \\
&= -\mathbf{w}^\top \exp\left\{ \widetilde{\mathbf{X}}\boldsymbol{\beta} + \widetilde{\mathbf{A}}\boldsymbol{\mu} + \frac{1}{2} \mathrm{diag}\left( \mathbf{A}\boldsymbol{\Sigma}\mathbf{A}^\top \right) \right\} + \mathbf{1}_N^\top (\mathbf{X}\boldsymbol{\beta} + \mathbf{A}\boldsymbol{\mu}).
\end{aligned} \tag{23}
$$

where the vector $\widetilde{\mathbf{A}}_i$ represents the $i$-th row of the matrix $\widetilde{\mathbf{A}}$.

For the derivation of the second part, we perform the Cholesky decomposition on the matrix $\boldsymbol{\Sigma}$ to get $\boldsymbol{\Sigma} = \mathbf{L}\mathbf{L}^\top$. Thus, $\mathbf{Z}$ can be expressed as $\mathbf{Z} = \boldsymbol{\mu} + \mathbf{L}\mathbf{x}$, where $\mathbf{x} \sim N(\mathbf{0}, \mathbf{I}_{n+N})$. Therefore, we can obtain:

$$
\begin{aligned}
\mathrm{E}_{q(\mathbf{Z})} \left[ \log p(\mathbf{Z}) \right] &= \mathrm{E}_{N(\mathbf{Z} \mid \boldsymbol{\mu}, \boldsymbol{\Sigma})} \left[ \log N(\mathbf{Z} \mid \mathbf{0}, \mathbf{K}_{\boldsymbol{\theta}}) \right] \\
&= \mathrm{E}_{N(\mathbf{Z} \mid \boldsymbol{\mu}, \boldsymbol{\Sigma})} \left( -\frac{1}{2} \log |\mathbf{K}_{\boldsymbol{\theta}}| - \frac{1}{2} \mathbf{Z}^\top \mathbf{K}_{\boldsymbol{\theta}}^{-1} \mathbf{Z} \right) \\
&= -\frac{1}{2} \log |\mathbf{K}_{\boldsymbol{\theta}}| - \frac{1}{2} \mathrm{E}_{N(\mathbf{x} \mid \mathbf{0}, \mathbf{I}_{n+N})} \left[ (\boldsymbol{\mu} + \mathbf{L}\mathbf{x})^\top \mathbf{K}_{\boldsymbol{\theta}}^{-1} (\boldsymbol{\mu} + \mathbf{L}\mathbf{x}) \right] \\
&= -\frac{1}{2} \log |\mathbf{K}_{\boldsymbol{\theta}}| - \frac{1}{2} \boldsymbol{\mu}^\top \mathbf{K}_{\boldsymbol{\theta}}^{-1} \boldsymbol{\mu} - \frac{1}{2} \mathrm{E}_{N(\mathbf{x} \mid \mathbf{0}, \mathbf{I}_{n+N})} \left( \mathbf{x}^\top \mathbf{L}^\top \mathbf{K}_{\boldsymbol{\theta}}^{-1} \mathbf{L}\mathbf{x} \right) \\
&= -\frac{1}{2} \log |\mathbf{K}_{\boldsymbol{\theta}}| - \frac{1}{2} \boldsymbol{\mu}^\top \mathbf{K}_{\boldsymbol{\theta}}^{-1} \boldsymbol{\mu} - \frac{1}{2} \mathrm{tr}\left( \mathbf{L}^\top \mathbf{K}_{\boldsymbol{\theta}}^{-1} \mathbf{L} \right) \\
&= -\frac{1}{2} \log |\mathbf{K}_{\boldsymbol{\theta}}| - \frac{1}{2} \boldsymbol{\mu}^\top \mathbf{K}_{\boldsymbol{\theta}}^{-1} \boldsymbol{\mu} - \frac{1}{2} \mathrm{tr}\left( \mathbf{K}_{\boldsymbol{\theta}}^{-1} \boldsymbol{\Sigma} \right).
\end{aligned} \tag{24}
$$

$$
E_{q(\mathbf{Z})} \left[ \log p\left(\mathbf{Z}\right) \right] = E_{N(\mathbf{Z}|\boldsymbol{\mu},\boldsymbol{\Sigma})} \left[ \log N\left(\mathbf{Z} \mid \boldsymbol{\mu}, \boldsymbol{\Sigma}\right) \right]
$$

$$
= E_{N(\mathbf{Z}|\boldsymbol{\mu},\boldsymbol{\Sigma})} \left[ -\frac{1}{2} \log |\boldsymbol{\Sigma}| - \frac{1}{2} \left(\mathbf{Z} - \boldsymbol{\mu}\right)^{\top} \mathbf{K}_{\boldsymbol{\theta}}^{-1} \left(\mathbf{Z} - \boldsymbol{\mu}\right) \right]
$$

$$
= -\frac{1}{2} \log |\boldsymbol{\Sigma}| - \frac{1}{2} E_{N(\mathbf{x}|\mathbf{0},\mathbf{I}_{n+N})} \left[ \left(\boldsymbol{\mu} + \mathbf{L}\mathbf{x} - \boldsymbol{\mu}\right)^{\top} \boldsymbol{\Sigma}^{-1} \left(\boldsymbol{\mu} + \mathbf{L}\mathbf{x} - \boldsymbol{\mu}\right) \right]
$$

$$
= -\frac{1}{2} \log |\boldsymbol{\Sigma}| - \frac{1}{2} E_{N(\mathbf{x}|\mathbf{0},\mathbf{I}_{n+N})} \left( \mathbf{x}^{\top} \mathbf{L}^{\top} \boldsymbol{\Sigma}^{-1} \mathbf{L}\mathbf{x} \right) \tag{25}
$$

$$
= -\frac{1}{2} \log |\boldsymbol{\Sigma}| - \frac{1}{2} \mathrm{tr}\left( \mathbf{L}^{\top} \boldsymbol{\Sigma}^{-1} \mathbf{L} \right)
$$

$$
= -\frac{1}{2} \log |\boldsymbol{\Sigma}| - \frac{1}{2} \mathrm{tr}\left( \boldsymbol{\Sigma}^{-1} \boldsymbol{\Sigma} \right)
$$

$$
= -\frac{1}{2} \log |\boldsymbol{\Sigma}| - \frac{n + N}{2}.
$$

Substituting (23), (24) and (25) into (22), we can obtain the explicit expression of ELBO as shown in the equation (6).

Next, in the following Lemma, we prove that the third line in the explicit expression of $E\left(\boldsymbol{\beta}, \boldsymbol{\mu}, \boldsymbol{\Sigma}\right)$ in (6) is equivalent to the KL divergence between two Gaussian distributions with the same mean but different covariance matrices.

**Lemma 1.** *The expression for the KL divergence between two Gaussian distributions with the same mean $f(\mathbf{x}) = N\left(\mathbf{x} \mid \boldsymbol{\mu}_m, \boldsymbol{\Sigma}\right)$ and $f_0(\mathbf{x}) = N\left(\mathbf{x} \mid \boldsymbol{\mu}_m, \mathbf{K}_{\boldsymbol{\theta}}\right)$ is*

$$
D\left(\boldsymbol{\Sigma}, \mathbf{K}_{\boldsymbol{\theta}}\right) = \frac{1}{2} \mathrm{tr}\left( \mathbf{K}_{\boldsymbol{\theta}}^{-1} \boldsymbol{\Sigma} \right) - \frac{1}{2} \log |\boldsymbol{\Sigma}| + \frac{1}{2} \log |\mathbf{K}_{\boldsymbol{\theta}}| - \frac{n + N}{2}. \tag{26}
$$

*Proof.* According to the definition of KL divergence, we can obtain

$$
D\left(\boldsymbol{\Sigma}, \mathbf{K}_{\boldsymbol{\theta}}\right) = \int f\left(\mathbf{x}\right) \log \frac{f\left(\mathbf{x}\right)}{f_0\left(\mathbf{x}\right)} \, \mathrm{d}\mathbf{x}
$$

$$
= \int f\left(\mathbf{x}\right) \log f\left(\mathbf{x}\right) \, \mathrm{d}\mathbf{x} - \int f\left(\mathbf{x}\right) \log f_0\left(\mathbf{x}\right) \, \mathrm{d}\mathbf{x}.
$$

$$
= E_{f(\mathbf{x})} \left[ \log f\left(\mathbf{x}\right) \right] - E_{f(\mathbf{x})} \left[ \log f_0\left(\mathbf{x}\right) \right]
$$

$$
= E_{N(\mathbf{x}|\boldsymbol{\mu}_m,\boldsymbol{\Sigma})} \left[ \log N\left(\mathbf{x} \mid \boldsymbol{\mu}_m, \boldsymbol{\Sigma}\right) \right] - E_{N(\mathbf{x}|\boldsymbol{\mu}_m,\boldsymbol{\Sigma})} \left[ \log N\left(\mathbf{x} \mid \boldsymbol{\mu}_m, \mathbf{K}_{\boldsymbol{\theta}}\right) \right] \tag{27}
$$

Based on the derivation of (25), we can get

$$
E_{N(\mathbf{x}|\boldsymbol{\mu}_m,\boldsymbol{\Sigma})} \left[ \log N\left(\mathbf{x} \mid \boldsymbol{\mu}_m, \boldsymbol{\Sigma}\right) \right] = -\frac{1}{2} \log |\boldsymbol{\Sigma}|
$$

$$
-\frac{1}{2} \left(\boldsymbol{\mu}_m - \boldsymbol{\mu}_m\right)^{\top} \mathbf{K}_{\boldsymbol{\theta}}^{-1} \left(\boldsymbol{\mu}_m - \boldsymbol{\mu}_m\right) - \frac{n}{2}, \tag{28}
$$

$$
E_{N(\mathbf{x}|\boldsymbol{\mu}_m,\boldsymbol{\Sigma})} \left[ \log N\left(\mathbf{x} \mid \boldsymbol{\mu}_m, \mathbf{K}_{\boldsymbol{\theta}}\right) \right] = -\frac{1}{2} \log |\mathbf{K}_{\boldsymbol{\theta}}|
$$

$$
-\frac{1}{2} \left(\boldsymbol{\mu} - \boldsymbol{\mu}_m\right)^{\top} \mathbf{K}_{\boldsymbol{\theta}}^{-1} \left(\boldsymbol{\mu} - \boldsymbol{\mu}_m\right) - \frac{1}{2} \mathrm{tr}\left( \mathbf{K}_{\boldsymbol{\theta}}^{-1} \boldsymbol{\Sigma} \right). \tag{29}
$$

Substituting the expressions (28) and (29) into (27) completes the proof. $\square$

## B   THE EXISTENCE AND UNIQUENESS OF THE OPTIMAL SOLUTION OF ELBO

Here, we prove Theorem 2.

To ensure the existence of the maximum value, we need to prove that $E\left(\boldsymbol{\beta}, \boldsymbol{\mu}, \boldsymbol{\Sigma}\right)$ is coercive on the convex set $\mathrm{R}^m \times \mathrm{R}^n \times \mathcal{S}_{n+N}^+$, that is, to prove that when any one of the parameters in $(\boldsymbol{\beta}, \boldsymbol{\mu}, \boldsymbol{\Sigma})$ approaches the boundary of its domain or infinity, the value of the function $E\left(\boldsymbol{\beta}, \boldsymbol{\mu}, \boldsymbol{\Sigma}\right) \to -\infty$. Next, we discuss the three cases corresponding to the three parameters respectively.

For $\boldsymbol{\mu}$, the expression of the term related to $\boldsymbol{\mu}$ in the function $E\left(\boldsymbol{\beta}, \boldsymbol{\mu}, \boldsymbol{\Sigma}\right)$ is

$$
E_1(\boldsymbol{\mu}) = -\mathbf{w}^{\top} \exp\left( \widetilde{\mathbf{A}} \boldsymbol{\mu} \right) + \mathbf{1}_N^{\top} \mathbf{A} \boldsymbol{\mu} - \frac{1}{2} \boldsymbol{\mu}^{\top} \mathbf{K}_{\boldsymbol{\theta}}^{-1} \boldsymbol{\mu}.
$$

When $\|\boldsymbol{\mu}\|_2 \to \infty$, for the quadratic term, since the covariance matrix $\mathbf{K}_{\boldsymbol{\theta}}$ is positive definite, the quadratic term approaches $-\infty$. Because the decay rate of the quadratic term is faster than that of the linear term, the sum of the quadratic term and the linear term must approach $-\infty$. For the exponential term $-\mathbf{w}^\top \exp\left(\widetilde{\mathbf{A}}\boldsymbol{\mu}\right)$, if one component in $\widetilde{\mathbf{A}}\boldsymbol{\mu}$ approaches $-\infty$, the exponential term approaches $0$; if one component in $\widetilde{\mathbf{A}}\boldsymbol{\mu}$ approaches $+\infty$, the exponential term approaches $-\infty$. Therefore, in summary, when $\|\boldsymbol{\mu}\|_2 \to \infty$, the value of the function $E_1(\boldsymbol{\mu})$ approaches $-\infty$.

For $\boldsymbol{\Sigma}$, the expression of the term related to $\boldsymbol{\Sigma}$ in the function $E\left(\boldsymbol{\beta}, \boldsymbol{\mu}, \boldsymbol{\Sigma}\right)$ is

$$E_2(\boldsymbol{\Sigma}) = -\mathbf{w}^\top \exp\left\{\frac{1}{2}\mathrm{diag}\left(\widetilde{\mathbf{A}}\boldsymbol{\Sigma}\widetilde{\mathbf{A}}^\top\right)\right\} - \frac{1}{2}\mathrm{tr}\left(\mathbf{K}_{\boldsymbol{\theta}}^{-1}\boldsymbol{\Sigma}\right) + \frac{1}{2}\log|\boldsymbol{\Sigma}|.$$

When $\boldsymbol{\Sigma}$ approaches the boundary of $\mathcal{S}_{n+N}^+$, that is, $\boldsymbol{\Sigma}$ becomes singular, which is equivalent to at least one eigenvalue of $\boldsymbol{\Sigma}$ satisfying $\lambda_i(\boldsymbol{\Sigma}) \to 0^+$. In this case, since $\log|\boldsymbol{\Sigma}| = \sum \log \lambda_i(\boldsymbol{\Sigma}) \to -\infty$, and both the exponential term and the linear term are negative, the value of the function $E_2(\boldsymbol{\Sigma})$ approaches $-\infty$; When $\boldsymbol{\Sigma}$ approaches "infinity", for example, when the maximum eigenvalue satisfies $\lambda_{\max}(\boldsymbol{\Sigma}) \to \infty$, the linear term $-\frac{1}{2}\mathrm{tr}\left(\mathbf{K}_{\boldsymbol{\theta}}^{-1}\boldsymbol{\Sigma}\right)$ approaches $-\infty$. Because the decay rate of the linear term is faster than the growth rate of the logarithmic term, and the exponential term is always negative, the value of the function $E_2(\boldsymbol{\Sigma})$ approaches $-\infty$.

For $\boldsymbol{\beta}$, the expression of the term related to $\boldsymbol{\beta}$ in the function $E\left(\boldsymbol{\beta}, \boldsymbol{\mu}, \boldsymbol{\Sigma}\right)$ is

$$E_3(\boldsymbol{\beta}) = -\mathbf{w}^\top \exp\left(\widetilde{\mathbf{X}}\boldsymbol{\beta}\right) + \mathbf{1}_N^\top\left(\mathbf{X}\boldsymbol{\beta}\right).$$

When $\|\boldsymbol{\beta}\|_2 \to \infty$, for the exponential term, since $\widetilde{\mathbf{X}}$ is the value of the covariate at $n$ integration points, in practice, a positive value of one component of $\widetilde{\mathbf{X}}\boldsymbol{\beta}$ can definitely be found. At this time, the exponential term approaches $-\infty$. Because the decay rate of the exponential term is faster than that of the linear term, when $\|\boldsymbol{\beta}\|_2 \to \infty$, the value of the function $E_3(\boldsymbol{\beta})$ approaches $-\infty$.

In conclusion, we have proved that the function $E\left(\boldsymbol{\beta}, \boldsymbol{\mu}, \boldsymbol{\Sigma}\right)$ is a coercive function on the convex set, which is equivalent to proving that $E\left(\boldsymbol{\beta}, \boldsymbol{\mu}, \boldsymbol{\Sigma}\right)$ has a maximum value on the convex set. According to the fact that $E\left(\boldsymbol{\beta}, \boldsymbol{\mu}, \boldsymbol{\Sigma}\right)$ is strongly concave obtained in Theorem 1, the maximum point is unique at this time.

## C  DERIVATION DETAILS AND THEORETICAL PROPERTY OF THE COORDINATE ASCENT MAXIMIZATION ALGORITHM

### C.1  DERIVATION OF $\boldsymbol{\Gamma}$ BASED ON NNGP

For the convenience, we abbreviate the $\mathbf{Z} = [Z(\widetilde{\mathbf{s}}_1), \ldots, Z(\widetilde{\mathbf{s}}_n), Z(\mathbf{s}_1), \ldots, Z(\mathbf{s}_N)]^\top$ as $\mathbf{Z} = [Z_1, \ldots Z_{n+N}]^\top$. Next, we can express the joint distribution of $\mathbf{Z}$ as

$$p(\mathbf{Z}) = p(Z_1) \prod_{i=2}^{n+N} p\left(Z_i \mid Z_{1:i-1}\right).$$

In NNGP, we let the realization of $Z(\mathbf{s})$ at any $\mathbf{s}_i \in \mathcal{I} \cup \mathcal{D}$ be conditional on, at most, the realizations at a pre-specified number of nearest neighbor locations to $\mathbf{s}_i$ in $\mathcal{I} \cup \mathcal{D}$, and this set is denoted as $\mathrm{Pa}[\mathbf{s}_i]$. Then, the approximate joint distribution can be expressed as

$$\widetilde{p}(\mathbf{Z}) = p(Z_1) \prod_{i=2}^{n+N} p\left(Z_i \mid Z_{\mathrm{Pa}[\mathbf{s}_i]}\right).$$

Datta et al. (2016) proved that $\widetilde{p}(\mathbf{Z})$ is the joint density of a multivariate Gaussian distribution with a sparse precision matrix, which satisfies

$$\sigma^{-2}\boldsymbol{\Gamma} = (\mathbf{I} - \mathbf{A})^\top \mathbf{D}^{-1}(\mathbf{I} - \mathbf{A}), \quad \widetilde{p}(\mathbf{Z}) = N\left(\mathbf{Z} \mid \mathbf{0}, \sigma^2\boldsymbol{\Gamma}\right).$$

where $\mathbf{I}$ is the $(n+N)$-dimensional identity matrix. The matrix $\mathbf{A}$ describes the neighbor information of all $n+N$ locations. The $i$-th row $(i > 1)$ of $\mathbf{A}$ has non-zero entries at the positions indexed by $\mathrm{Pa}[\mathbf{s}_i]$, and these non-zero entries are calculated as follows

$$\mathbf{A}\left(i, \mathrm{Pa}[\mathbf{s}_i]\right) = \mathbf{K}_{\boldsymbol{\theta}}\left(\mathbf{s}_i, \mathrm{Pa}[\mathbf{s}_i]\right)\left(\mathbf{K}_{\boldsymbol{\theta}}\left(\mathrm{Pa}[\mathbf{s}_i], \mathrm{Pa}[\mathbf{s}_i]\right)\right)^{-1}.$$

The matrix $\mathbf{D}$ is a diagonal matrix, and its $i$-th diagonal element describes the variance information at $\mathbf{s}_i$, which satisfies

$$\mathbf{D}(i,i) = \mathbf{K}_{\boldsymbol{\theta}}(\mathbf{s}_i,\mathbf{s}_i) - \mathbf{K}_{\boldsymbol{\theta}}\left(\mathbf{s}_i, \mathrm{Pa}[\mathbf{s}_i]\right) \left(\mathbf{K}_{\boldsymbol{\theta}}\left(\mathrm{Pa}[\mathbf{s}_i], \mathrm{Pa}[\mathbf{s}_i]\right)\right)^{-1} \mathbf{K}_{\boldsymbol{\theta}}\left(\mathrm{Pa}[\mathbf{s}_i], \mathbf{s}_i\right).$$

Finley et al. (2019) discussed methods for efficiently constructing $\mathbf{A}$ and $\mathbf{D}$. On this basis, Zhang et al. (2019) implemented the above construction in practice based on `rStan` in R.

## C.2 THE DERIVATION DETAILS OF THE EXPRESSION OF $\nabla_{\boldsymbol{\Sigma}}E$

**Lemma 2.** *The gradient of the objective function $E_{\boldsymbol{\theta}}\left(\boldsymbol{\beta}, \boldsymbol{\mu}, \boldsymbol{\Sigma}\right)$ with respect to $\boldsymbol{\Sigma}$ is*

$$\nabla_{\boldsymbol{\Sigma}}E = \frac{1}{2}\left[-\widetilde{\mathbf{A}}^{\top}\mathrm{Diag}\left[\mathbf{w}\circ\exp\left\{\widetilde{\mathbf{X}}\boldsymbol{\beta} + \widetilde{\mathbf{A}}\boldsymbol{\mu} + \frac{1}{2}\mathrm{diag}\left(\widetilde{\mathbf{A}}\boldsymbol{\Sigma}\widetilde{\mathbf{A}}^{\top}\right)\right\}\right]\widetilde{\mathbf{A}} - \sigma^{-2}\boldsymbol{\Gamma} + \boldsymbol{\Sigma}^{-1}\right].$$

*Proof.* Define the matrix function $g(\boldsymbol{\Sigma}) = \widetilde{\mathbf{X}}\boldsymbol{\beta} + \widetilde{\mathbf{A}}\boldsymbol{\mu} + \frac{1}{2}\mathrm{diag}\left(\widetilde{\mathbf{A}}\boldsymbol{\Sigma}\widetilde{\mathbf{A}}^{\top}\right)$, where the $i$-th component of $g(\boldsymbol{\Sigma})$ satisfies

$$g_i(\boldsymbol{\Sigma}) = \left(\widetilde{\mathbf{X}}\boldsymbol{\beta} + \widetilde{\mathbf{A}}\boldsymbol{\mu}\right)_i + \frac{1}{2}\left[\mathrm{diag}\left(\widetilde{\mathbf{A}}\boldsymbol{\Sigma}\widetilde{\mathbf{A}}^{\top}\right)\right]_i = \left(\widetilde{\mathbf{X}}\boldsymbol{\beta} + \widetilde{\mathbf{A}}\boldsymbol{\mu}\right)_i + \frac{1}{2}\left(\widetilde{\mathbf{A}}\boldsymbol{\Sigma}\widetilde{\mathbf{A}}^{\top}\right)_{ii}$$

Given that the expansion of the elements of the quadratic form matrix can be expressed as $\left(\widetilde{\mathbf{A}}\boldsymbol{\Sigma}\widetilde{\mathbf{A}}^{\top}\right)_{ii} = \sum_{j,k}\widetilde{\mathbf{A}}_{ij}\boldsymbol{\Sigma}_{jk}\widetilde{\mathbf{A}}_{ik}$, thus the partial derivative of $g_i(\boldsymbol{\Sigma})$ with respect to the matrix elements can be obtained as

$$\frac{\partial\left(\widetilde{\mathbf{A}}\boldsymbol{\Sigma}\widetilde{\mathbf{A}}^{\top}\right)_{ii}}{\partial\boldsymbol{\Sigma}_{jk}} = \frac{\partial\left(\widetilde{\mathbf{A}}\boldsymbol{\Sigma}\widetilde{\mathbf{A}}^{\top}\right)_{ii}}{\partial\boldsymbol{\Sigma}_{jk}} = \frac{1}{2}\widetilde{\mathbf{A}}_{ij}\widetilde{\mathbf{A}}_{ik}$$

Writing it in matrix form is equivalent to

$$\frac{\partial g_i(\boldsymbol{\Sigma})}{\partial\boldsymbol{\Sigma}} = \frac{1}{2}\widetilde{\mathbf{A}}_i^{\top}\widetilde{\mathbf{A}}_i$$

where $\widetilde{\mathbf{A}}_i$ represents the $i$-th row vector of the matrix $\widetilde{\mathbf{A}}$.

Based on the chain rule, the gradient of the first term with respect to the matrix $\boldsymbol{\Sigma}$ can be derived as follows

$$\begin{aligned}
\frac{\partial}{\partial\boldsymbol{\Sigma}}\left(-\mathbf{w}^{\top}\exp\{g(\boldsymbol{\Sigma})\}\right) &= -\sum_{i=1}^{N}w_i\frac{\partial}{\partial\boldsymbol{\Sigma}}\exp\left\{g_i(\boldsymbol{\Sigma})\right\} \\
&= -\sum_{i=1}^{N}w_i\exp\left\{g_i(\boldsymbol{\Sigma})\right\}\frac{\partial g_i(\boldsymbol{\Sigma})}{\partial\boldsymbol{\Sigma}} \\
&= -\sum_{i=1}^{N}w_i\exp\left\{\left(\widetilde{\mathbf{X}}\boldsymbol{\beta} + \widetilde{\mathbf{A}}\boldsymbol{\mu}\right)_i + \frac{1}{2}\left(\widetilde{\mathbf{A}}\boldsymbol{\Sigma}\widetilde{\mathbf{A}}^{\top}\right)_{ii}\right\}\cdot\frac{1}{2}\widetilde{\mathbf{A}}_i^{\top}\widetilde{\mathbf{A}}_i \\
&= -\frac{1}{2}\sum_{i=1}^{N}w_i\exp\left\{\left(\widetilde{\mathbf{X}}\boldsymbol{\beta} + \widetilde{\mathbf{A}}\boldsymbol{\mu}\right)_i + \frac{1}{2}\left(\widetilde{\mathbf{A}}\boldsymbol{\Sigma}\widetilde{\mathbf{A}}^{\top}\right)_{ii}\right\}\widetilde{\mathbf{A}}_i^{\top}\widetilde{\mathbf{A}}_i \\
&\triangleq -\frac{1}{2}\widetilde{\mathbf{A}}^{\top}\mathrm{Diag}\left[\mathbf{w}\circ\exp\left\{\widetilde{\mathbf{X}}\boldsymbol{\beta} + \widetilde{\mathbf{A}}\boldsymbol{\mu} + \frac{1}{2}\mathrm{diag}\left(\widetilde{\mathbf{A}}\boldsymbol{\Sigma}\widetilde{\mathbf{A}}^{\top}\right)\right\}\right]\widetilde{\mathbf{A}}.
\end{aligned}$$

For the gradients of the second and third terms with respect to the matrix $\Sigma$, according to the following properties

$$\frac{\partial}{\partial\boldsymbol{\Sigma}}\mathrm{tr}(\boldsymbol{\Gamma}\boldsymbol{\Sigma}) = \boldsymbol{\Gamma}^{\top}, \quad \frac{\partial}{\partial\boldsymbol{\Sigma}}\log|\boldsymbol{\Sigma}| = (\boldsymbol{\Sigma}^{-1})^{\top},$$

and since both $\boldsymbol{\Gamma}$ and $\boldsymbol{\Sigma}$ are symmetric, thus $\boldsymbol{\Gamma}^{\top} = \boldsymbol{\Gamma}$, $(\boldsymbol{\Sigma}^{-1})^{\top} = \boldsymbol{\Sigma}^{-1}$.

By organizing the gradient results of the three parts with respect to the matrix $\Sigma$, the result of Lemma 2 can be obtained. $\qquad\square$

### C.3 THE CONVERGENCE ANALYSIS OF FIXED-POINT METHOD FOR UPDATING $\mathbf{\Sigma}$

In this paragraph, we first prove in Theorem 4 that the sequence $\{\mathbf{\Sigma}_k\}_{k \geq 0}$ obtained based on the fixed-point method (15) converges.

**Theorem 4.** *The sequence $\{\mathbf{\Sigma}_k\}_{k \geq 0}$ obtained by iteration based on the fixed-point method is convergent.*

*Proof.* According to the iterative relationship of the fixed-point method (15), for any $k = 1, 2, \ldots$, we have $\mathbf{\Sigma}_k \preceq \sigma^2 \mathbf{\Gamma}^{-1}$, which is equivalent to $\|\mathbf{\Sigma}_k\|_2 \leq \|\sigma^2 \mathbf{\Gamma}^{-1}\|_2$. Therefore, the sequence of matrix variables $\{\mathbf{\Sigma}_k\}$ is bounded in the spectral norm. Since every bounded sequence has a convergent subsequence, there exists a subsequence $\{\mathbf{\Sigma}_{(n)}\}_{n \geq 0}$ such that it converges to the limit $\mathbf{\Sigma}_\star$ in the sense of the 2-norm. $\qquad\square$

Furthermore, we give the specific form of the above convergent subsequence in Lemma 3 and Theorem 5.

For any positive definite matrices $\mathbf{C}, \mathbf{D} \in \mathcal{S}_{n+N}^+$, assume that $\mathbf{C} \preceq \mathbf{D}$, if the mapping $T$ satisfies $T(\mathbf{C}) \succeq T(\mathbf{D})$, then the mapping $T$ is said to be anti-monotonic.

**Lemma 3.** *The mapping $F$ corresponding to the fixed-point method* (15) *is given by*

$$F(\mathbf{\Sigma}) = \left[ \widetilde{\mathbf{A}}^\top \mathrm{Diag} \left[ \mathbf{w} \circ \exp \left\{ \widetilde{\mathbf{X}}\boldsymbol{\beta} + \widetilde{\mathbf{A}}\boldsymbol{\mu} + \frac{1}{2} \mathrm{diag}\left( \widetilde{\mathbf{A}}\mathbf{\Sigma}\widetilde{\mathbf{A}}^\top \right) \right\} \right] \widetilde{\mathbf{A}} + \sigma^{-2}\mathbf{\Gamma} \right]^{-1}. \qquad (30)$$

*Then, the mapping $F$ is anti-monotonic.*

*Proof.* The following proof refers to (Arridge et al., 2018, Appendix A.). Given two positive definite matrices $\mathbf{\Sigma}_1$ and $\mathbf{\Sigma}_2$, assume that $\mathbf{\Sigma}_1 \preceq \mathbf{\Sigma}_2$. Then, each component value of the vector $\mathrm{diag}(\widetilde{\mathbf{A}}\mathbf{\Sigma}_1\widetilde{\mathbf{A}}^\top)$ is less than the value of the corresponding position of the vector $\mathrm{diag}(\widetilde{\mathbf{A}}\mathbf{\Sigma}_2\widetilde{\mathbf{A}}^\top)$, denoted as $\mathrm{diag}(\widetilde{\mathbf{A}}\mathbf{\Sigma}_1\widetilde{\mathbf{A}}^\top) \leq \mathrm{diag}(\widetilde{\mathbf{A}}\mathbf{\Sigma}_2\widetilde{\mathbf{A}}^\top)$. At this time, we can obtain

$$T(\mathbf{\Sigma}_1) - T(\mathbf{\Sigma}_2) = T(\mathbf{\Sigma}_1) \left\{ (T(\mathbf{\Sigma}_2))^{-1} - (T(\mathbf{\Sigma}_1))^{-1} \right\} T(\mathbf{\Sigma}_2) \succeq \mathbf{0}.$$

Therefore, the lemma is proved. $\qquad\square$

Next, we construct two convergent subsequences of the maximization sequence $\{\mathbf{\Sigma}_k\}_{k \geq 0}$ obtained by the fixed-point method (15).

**Theorem 5.** *Given any initial iteration value $\mathbf{\Sigma}_0$, the maximization sequence $\{\mathbf{\Sigma}_k\}_{k \geq 0}$ has two convergent subsequences $\{\mathbf{\Sigma}_{2k}\}_{k \geq 0}$ and $\{\mathbf{\Sigma}_{2k+1}\}_{k \geq 0}$.*

*Proof.* The following proof refers to (Arridge et al., 2018, Appendix A.). According to the expression of the fixed - point method (15), for any $k \geq 0$, we have

$$\mathbf{0} \preceq \mathbf{\Sigma}_k \preceq \mathbf{\Sigma}_0,$$

so the maximization sequence $\{\mathbf{\Sigma}_k\}_{k \geq 0}$ is always bounded. Based on the iterative relation of the fixed - point method (15) and the conclusion of the anti - monotonicity of $F$ obtained from Lemma 3, if $\mathbf{\Sigma}_i \succeq \mathbf{\Sigma}_j$, then $\mathbf{\Sigma}_{i+1} \preceq \mathbf{\Sigma}_{j+1}$; similarly, if $\mathbf{\Sigma}_i \preceq \mathbf{\Sigma}_j$, then $\mathbf{\Sigma}_{i+1} \succeq \mathbf{\Sigma}_{j+1}$. Therefore, in the sense of the 2 - norm, the sequence $\{\mathbf{\Sigma}_{2k}\}_{k \geq 0}$ is a decreasing sequence, and the sequence $\{\mathbf{\Sigma}_{2k+1}\}_{k \geq 0}$ is an increasing sequence. Since a monotonic and bounded sequence must converge, $\{\mathbf{\Sigma}_{2k}\}_{k \geq 0}$ and $\{\mathbf{\Sigma}_{2k+1}\}_{k \geq 0}$ are two convergent subsequences of the maximization sequence $\{\mathbf{\Sigma}_k\}_{k \geq 0}$, and both are the limits of the fixed-point mapping $F^2$. $\qquad\square$

### C.4 THE CONVERGENCE ANALYSIS OF ALGORITHM 1

For the maximizing sequence $\{\boldsymbol{\beta}_k\}_{k\geq 0}$ and $\{\boldsymbol{\mu}_k\}_{k\geq 0}$ obtained by the Newton method iteration. From the expression of the Hessian matrices (12), it is easy to see that $-\nabla_{\boldsymbol{\beta}}^2 E$ and $-\nabla_{\boldsymbol{\mu}}^2 E$ are strictly positive definite. Consequently, the Newton method for these updates is guaranteed to converge to the unique optimum of each subproblem (Kelley, 1995). For the maximizing sequence $\{\boldsymbol{\beta}_k\}_{k\geq 0}$ obtained by the fixed - point method, Theorem 4 guarantees the convergence of the sequence.

Finally, according to the convergence criterion of the coordinate ascent method given in (Bertsekas, 1997, proposition 2.7.1), if for each coordinate direction, there exists a unique point that maximizes the objective function in the current coordinate direction, then the coordinate sequence obtained by the coordinate ascent method is convergent. In the current context, based on the joint strong concavity of $E(\boldsymbol{\beta}, \boldsymbol{\mu}, \boldsymbol{\Sigma})$ with respect to $(\boldsymbol{\beta}, \boldsymbol{\mu}, \boldsymbol{\Sigma})$ obtained from Theorem 1, and the convergence of the coordinate sequences obtained by the Newton method and the fixed - point method in their respective directions, the sequences $\{\boldsymbol{\beta}_k\}_{k\geq 0}, \{\boldsymbol{\mu}_k\}_{k\geq 0}, \{\boldsymbol{\Sigma}_k\}_{k\geq 0}$ obtained by Algorithm 1 are convergent.

## D ANALYSIS OF THE CONVERGENCE OF THE HYPERPARAMETER MONOTONIC CONVERGENCE ALGORITHM

To analyze the convergence of Algorithm 2, that is, to prove Theorem 3, we first need to introduce the following lemmas.

For convenience, when $\sigma$ is fixed, we denote the optimal parameter values obtained based on Algorithm 1 as $\boldsymbol{\beta}_\sigma$, $\boldsymbol{\mu}_\sigma$ and $\boldsymbol{\Sigma}_\sigma$. During the iterative process, since only the hyperparameter $\sigma$ in $E_{\boldsymbol{\theta}}(\boldsymbol{\beta}, \boldsymbol{\mu}, \boldsymbol{\Sigma})$ changes, we will denote $E_{\boldsymbol{\theta}}(\boldsymbol{\beta}, \boldsymbol{\mu}, \boldsymbol{\Sigma})$ as $E_{\sigma}(\boldsymbol{\beta}, \boldsymbol{\mu}, \boldsymbol{\Sigma})$ in the following.

Next, we further decompose the function $E_{\boldsymbol{\theta}}(\boldsymbol{\beta}, \boldsymbol{\mu}, \boldsymbol{\Sigma})$ as follows:

$$E_{\sigma}(\boldsymbol{\beta}, \boldsymbol{\mu}, \boldsymbol{\Sigma}) = g(\boldsymbol{\beta}, \boldsymbol{\mu}, \boldsymbol{\Sigma}) + \sigma^{-2} h(\boldsymbol{\mu}, \boldsymbol{\Sigma}) + \frac{n+N}{2} \log \sigma^{-2},$$

where

$$g(\boldsymbol{\beta}, \boldsymbol{\mu}, \boldsymbol{\Sigma}) = -\mathbf{w}^\top \exp\left\{ \widetilde{\mathbf{X}}\boldsymbol{\beta} + \widetilde{\mathbf{A}}\boldsymbol{\mu} + \frac{1}{2}\mathrm{diag}\left( \widetilde{\mathbf{A}}\boldsymbol{\Sigma}\widetilde{\mathbf{A}}^\top \right) \right\}$$

$$+ \mathbf{1}_N^\top (\mathbf{X}\boldsymbol{\beta} + \mathbf{A}\boldsymbol{\mu}) + \frac{1}{2}\log|\boldsymbol{\Sigma}| - \frac{1}{2}\log|\boldsymbol{\Gamma}^{-1}|,$$

$$h(\boldsymbol{\mu}, \boldsymbol{\Sigma}) = -\frac{1}{2}\boldsymbol{\mu}^\top \boldsymbol{\Gamma}\boldsymbol{\mu} - \frac{1}{2}\mathrm{tr}(\boldsymbol{\Gamma}\boldsymbol{\Sigma}).$$

Next, we introduce the following lemma.

**Lemma 4.** *The function $h(\boldsymbol{\mu}_\sigma, \boldsymbol{\Sigma}_\sigma)$ is monotonically decreasing with respect to $\sigma$.*

*Proof.* The proof of this lemma refers to (Arridge et al., 2018, Lemma 5.3). Given any two positive numbers $\sigma_1$ and $\sigma_2$, according to Theorem 1 and Theorem 2, we can find the optimal values of the parameters $(\boldsymbol{\beta}, \boldsymbol{\mu}, \boldsymbol{\Sigma})$ for the currently fixed $\sigma$. When $\sigma$ is fixed to $\sigma_1$ and $\sigma_2$ respectively, we denote the optimal values of the parameters as $(\boldsymbol{\beta}_{\sigma_1}, \boldsymbol{\mu}_{\sigma_1}, \boldsymbol{\Sigma}_{\sigma_1})$ and $(\boldsymbol{\beta}_{\sigma_2}, \boldsymbol{\mu}_{\sigma_2}, \boldsymbol{\Sigma}_{\sigma_2})$ respectively. Thus, we can obtain

$$E_{\sigma_1}(\boldsymbol{\beta}_{\sigma_1}, \boldsymbol{\mu}_{\sigma_1}, \boldsymbol{\Sigma}_{\sigma_1}) \geq E_{\sigma_1}(\boldsymbol{\beta}_{\sigma_2}, \boldsymbol{\mu}_{\sigma_2}, \boldsymbol{\Sigma}_{\sigma_2}),$$
$$E_{\sigma_2}(\boldsymbol{\beta}_{\sigma_2}, \boldsymbol{\mu}_{\sigma_2}, \boldsymbol{\Sigma}_{\sigma_2}) \geq E_{\sigma_2}(\boldsymbol{\beta}_{\sigma_1}, \boldsymbol{\mu}_{\sigma_1}, \boldsymbol{\Sigma}_{\sigma_1}).$$

Adding the left - hand sides and right - hand sides of the above two inequalities respectively, and then continuing to simplify, we can get

$$\left( \sigma_1^{-2} - \sigma_2^{-2} \right) \left( h(\boldsymbol{\mu}_{\sigma_1}, \boldsymbol{\Sigma}_{\sigma_1}) - h(\boldsymbol{\mu}_{\sigma_2}, \boldsymbol{\Sigma}_{\sigma_2}) \right) \geq 0.$$

Therefore, it can be proved that $h(\boldsymbol{\mu}_\sigma, \boldsymbol{\Sigma}_\sigma)$ is monotonically increasing with respect to $\sigma^{-2}$. It should be noted that since $\sigma > 0$, this conclusion is equivalent to being monotonically decreasing with respect to $\sigma$. $\square$

Based on the conclusion of Lemma 4, we will prove Theorem 3, which refers to (Arridge et al., 2018, Theorem 5.1).

Based on the expression for updating $\sigma$ given in 21, for any $k \geq 1$, we can obtain

$$\sigma_{k+1}^2 - \sigma_k^2 = \frac{-h\left(\boldsymbol{\mu}_{\sigma_k}, \boldsymbol{\Sigma}_{\sigma_k}\right) + h\left(\boldsymbol{\mu}_{\sigma_{k-1}}, \boldsymbol{\Sigma}_{\sigma_{k-1}}\right)}{n + N + 2a + 2}$$

At this point, the magnitude relationship between $\sigma_{k+1}^2$ and $\sigma_k^2$ depends only on the magnitude relationship between the function values $h\left(\boldsymbol{\mu}_{\sigma_k}, \boldsymbol{\Sigma}_{\sigma_k}\right)$ and $h\left(\boldsymbol{\mu}_{\sigma_{k-1}}, \boldsymbol{\Sigma}_{\sigma_{k-1}}\right)$. Based on Lemma 4, if $\sigma_k^2 \geq \sigma_{k-1}^2$, then $h(\boldsymbol{\mu}_{\sigma_k}, \boldsymbol{\Sigma}_{\sigma_k}) \leq h(\boldsymbol{\mu}_{\sigma_{k-1}}, \boldsymbol{\Sigma}_{\sigma_{k-1}})$, which means $\sigma_{k+1}^2 \geq \sigma_k^2$. Similarly, if $\sigma_k^2 \leq \sigma_{k-1}^2$, then $\sigma_{k+1}^2 \leq \sigma_k^2$. Therefore, the magnitude relationship between $\sigma_{k+1}^2$ and $\sigma_k^2$ always remains the same as that between $\sigma_k^2$ and $\sigma_{k-1}^2$, indicating that the sequence $\{\sigma_k^2\}$ is monotonic. In addition, since $-h(\boldsymbol{\mu}, \boldsymbol{\Sigma}) > 0$, and according to the boundedness of the parameters given in Theorem 2, the sequence $\{\sigma_k^2\}_{k \geq 0}$ is bounded. Since a monotonic and bounded sequence must converge, there exists a $\sigma_\star^2$ such that $\lim_{k \to +\infty} \sigma_k^2 = \sigma_\star^2$. Since $\sigma_k > 0$, this is equivalent to $\lim_{k \to +\infty} \sigma_k = \sigma_\star$.

# E    NUMERICAL EXPERIMENTS DETAILS

In this section, we provide supplementary explanations for the simulation settings and the parameter settings related to the algorithm in Section 4. All numerical simulations and comparative experiments of different methods were completed on a Windows 11 laptop equipped with an Intel Core i7-10750H processor (2.60 GHz, 12 threads) and 16 GB of memory. Moreover, all the code was implemented in R language.

**LGCP point pattern simulation settings**    Based on the rLGCP() function in the spatstat package Baddeley & Turner (2005), we simulated a set of point patterns following LGCP within the square observation region $\Omega = [0, 10] \times [0, 10]$, and the expression of its intensity function is

$$\log \lambda(\mathbf{s}) = \beta_0 + X(\mathbf{s})\beta_1 + Z(\mathbf{s}),$$

where $X(\mathbf{s})$ is defined as a one-dimensional covariate with the expression $X(\mathbf{s}) = \cos(s_1 - 2.5) - \sin(s_2 - 3.5)$; the parameters $\beta_0$ and $\beta_1$ represent the intercept and the coefficient of the covariate respectively; For the last term $Z(\mathbf{s})$, we choose a Gaussian random field with mean $\mathbf{0}$ and a kernel function from the Matérn class. The expression of the Matérn kernel at $\mathbf{s}_1$ and $\mathbf{s}_2$ satisfies

$$K_{\boldsymbol{\theta}}(\mathbf{s}, \mathbf{s}') = \frac{\sigma^2}{2^{\nu-1}\Gamma(\nu)} \left(\kappa\|\mathbf{s} - \mathbf{s}'\|\right)^\nu K_\nu\left(\kappa\|\mathbf{s} - \mathbf{s}'\|\right), \tag{31}$$

where $\|\mathbf{s} - \mathbf{s}'\|$ is the Euclidean distance between $\mathbf{s}$ and $\mathbf{s}'$, $\sigma^2$ is the marginal variance, $\kappa > 0$ is the scale parameter, and $\nu > 0$ is the smoothness parameter, $K_\nu(\cdot)$ is the modified Bessel function of the second kind. We set the intercept and covariate coefficient to $(\beta_0, \beta_1) = (2.5, 0.8)$, and the hyperparameters of the Matérn random field to $\boldsymbol{\theta} = \left(\sigma^2, \kappa, \nu\right) = (0.22, 0.3, 1)$.

According to the above method, we generated the LGCP log-intensity surface as shown in Figure 2(a) on $\Omega$ and the observed points consisting of $N = 2289$ point patterns. In "full case", we retain all the observed points; in "hole case", we consider the issue of sampling effort, that is, in the original point pattern, some points cannot be observed. For the selection of these unobservable points, we divide it into two strategies. First, we specify that the area inside the red box is the unobservable area; second, on this basis, we randomly discard 20% of the remaining observed points as the unrecorded observed points. At this time, the number of remaining observed points is $N = 1219$. The "hole case" setting is designed to evaluate the model's ability to capture both the global and local characteristics of the LGCP intensity function.

**Parameter initialization settings in the VoGCAM algorithm**    In our method, we should first select $\mathbf{Z}(\mathbf{s})$ as the Matérn field defined in 31, which is widely used in spatial statistics to model random fields Stein (1999). Next, we discuss the selection of hyperparameters $\nu$ and $\kappa$. Empirically, the smoothness parameter $\nu$ is often difficult to estimate well due to multi-modality issues in spatial statistics (Finley et al., 2019). Therefore, $\nu$ is typically set to common values such as 0.5 or 1.5.

In our setting, we initialize it as $\nu = 0.5$. For estimating $\kappa$, we adopt the empirical estimation method proposed by Lindgren et al. (2011) by set $\kappa = \sqrt{8\nu/\rho}$, where $\rho$ is the range parameter. In practice, $\rho$ can be set to the maximum Euclidean distance between any two points within the current observation domain. Thus, by determining the values of $\nu$ and $\rho$, we can determine $\kappa$. In our setting, we choose $\rho = 10\sqrt{2}$, which is corresponding to $\kappa = 0.2$.

For more detailed parameter settings, please refer to the supplement file.

**Verification of Algorithmic Convergence.** Under the data pattern as shown in Figure 2 (b), we verify the convergence of the VoGCAM algorithm 2 by selecting two different sets of initial parameter values. Figure 4 illustrates the evolution of the target parameters and the ELBO across iterations for two different initializations. At iteration $k$, we quantify parameter updates by the Euclidean norms $\|\delta\boldsymbol{\beta}\|_2$ and $\|\delta\boldsymbol{\mu}\|_2$, and by the Frobenius norm $\|\delta\boldsymbol{\Sigma}\|_F$. The corresponding ELBO value is $E(\boldsymbol{\beta}_k, \boldsymbol{\mu}_k, \boldsymbol{\Sigma}_k; \sigma_k)$. In both experiments, all parameter sequences and the ELBO converge. Although the hyperparameter $\sigma_k$ typically approaches its limit more slowly than others, its variation have minimal impact on the ELBO. Hence, once the ELBO values stabilize, we consider the algorithm to be converged.

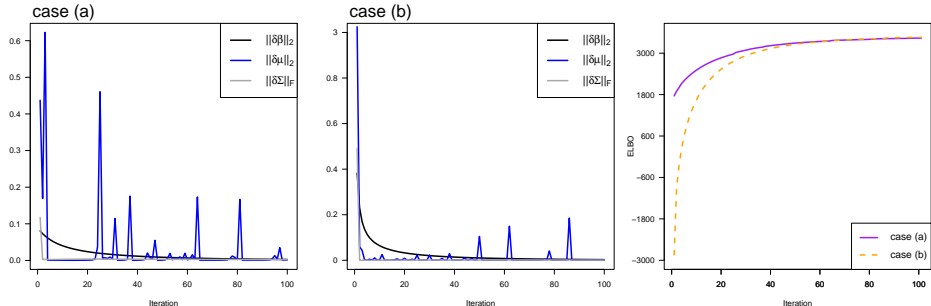

Figure 4: Case (a) and case (b) show the relative error of the target parameters changing with the number of iterations under two different sets of initial values. The relative error $\|\delta\mathbf{x}\|$ at the $k$-th step is defined as $\|\mathbf{x}_{k+1} - \mathbf{x}_k\|$. The rightmost figure shows the ELBO changing with the number of iterations under the initial values corresponding to case (a) and case (b).

# F LIMITATIONS

This paper introduces a method for fitting LGCP, which can be readily extended to more general Cox processes to model a wider range of point-pattern types. In the standard LGCP framework, all spatial heterogeneity is governed by a single intensity function, yielding a single-structure point process. In contrast, many real-world phenomena require multi-structure point processes that combine several spatiotemporal random components. For instance, when examining the spatial distributions of multiple tree species in a forest, each species may follow its own point process. To accommodate such complexities, the intensity function must be generalized to capture common multi-structure mechanisms, including hybridization, sparsification, and superposition. Performing variational inference in this multi-structure setting poses significant challenges, as the resulting objective functions generally lack closed-form expressions. Addressing these challenges will therefore require the development of novel modeling frameworks and inference techniques.

# G LLM USAGE

The authors used large language models (LLMs), specifically Gemini 2.5 and ChatGPT-5, as writing assistants to improve the clarity, grammar, and fluency of the English prose. The use of these tools was limited to refining existing content and did not involve generating core ideas, analytical insights, or experimental results. All content presented in this paper is the sole intellectual creation of the human authors, who take full responsibility for its accuracy and originality.

