# OpenReview forum: "An Efficient Variational Method for Fitting Log-Gaussian Cox Processes"
_ICLR.cc/2026/Conference — Submitted to ICLR 2026_

### Official Review · Reviewer_z2Fg · 2025-10-30

**Soundness:** 2
**Presentation:** 3
**Contribution:** 1
**Rating:** 2
**Confidence:** 4

**Summary:**

This paper proposes an efficient variational method for fitting log-Gaussian Cox processes (LGCPs). The intractable integral term in the LGCP likelihood is approximated using a Voronoi tessellation approach, and a variational Bayesian framework with a Gaussian proposal distribution is developed. The resulting evidence lower bound (ELBO) is proven to be strongly concave and to admit a unique maximizer. A coordinate ascent algorithm that combines Newton and fixed-point updates, together with a nearest-neighbor Gaussian process and the Woodbury matrix identity, ensures scalable model fitting with respect to both data size and the Voronoi discretization level. Experiments on simulated and real spatial data demonstrate that the proposed method achieves superior predictive accuracy compared with conventional alternatives.

**Strengths:**

- Since log-Gaussian Cox processes (LGCPs) are widely used models for analyzing spatial event data, a concave formulation for LGCP has significant practical value.
- The paper carefully develops an efficient algorithm for parameter and hyperparameter learning by effectively leveraging existing ideas.
- The manuscript is clearly written and easy to follow.

**Weaknesses:**

- In the literature of point processes, approximating the integral in the likelihood function by a summation has a long history. Common approaches include the use of uniform grids and Monte Carlo integration, and the Voronoi tessellation method can be regarded as a variant of these. As noted in the paper, this approach has some desirable convergence properties, but the idea itself is not particularly novel: it appears to be a straightforward application of existing techniques. If applying Voronoi tessellation to the log-Gaussian Cox process offers substantial advantages or unique insights, these should be clearly articulated.
- I appreciate the careful construction of an efficient algorithm for model fitting. However, the techniques employed (e.g., low-rank approximations of large matrices) are based on conventional ideas in GP literature and are not non-trivial in themselves.
- The positioning of the proposed method relative to prior research is ambiguous. In general, when considering spatial point process models, the problem formulation and objectives differ substantially depending on whether or not spatial covariates are assumed to exist. In the related work section, references to these two settings are discussed without a clear distinction, leaving the reader uncertain about how the authors intend to position their proposed model.
    - If the authors tackle the problem of estimating the relationship between covariates and the intensity function, then prior studies addressing this problem (e.g., [1,2,3]), as well as INLA, should also be reviewed and compared.
    - Alternatively, if the authors consider that spatial covariates are not mandatory, then the comparative discussion about efficient Gaussian Cox processes with the square [4,5] and sigmoidal (e.g., [6,7]) link functions is necessary. Especially, if the reason for focusing on the somewhat classical log-GCP lies in its high predictive accuracy, and the main contribution of this study is to further improve that accuracy, then it would be valuable to include several recent GCP models [4,5,6,7] (not necessarily exhaustively) as baselines and discuss differences in both predictive performance and computational efficiency.

[1] Baddely et al., “Nonparametric estimation of the dependence of a spatial point process on spatial covariates”, Statistics and Its Interface, 2012.

[2] Yu and Loh, “Bayesian semiparametric intensity estimation for inhomogeneous spatial point processes”, Biometrics, 2011.

[3] Kim et al., “Fast Bayesian estimation of point process intensity as function of covariates”, NeurIPS, 2022.

[4] Lloyd et al., “Variational Inference for Gaussian Process Modulated Poisson Processes”, ICML, 2015.

[5] John and Hensman, “Large-Scale Cox Process Inference using Variational Fourier Features”, ICML, 2018.

[6] Donner and Opper, “Efficient Bayesian Inference of Sigmoidal Gaussian Cox Processes”, JMLR, 2018.

[7] Aglietti et al., “Structured Variational Inference in Continuous Cox Process Models”, NeurIPS, 2019.

**Questions:**

see strengths/weaknesses.

---

> ### Author Response · Authors · 2025-11-22
>
> We thank the reviewer for their rigorous assessment and for providing detailed references regarding alternative link functions and covariate estimation. We appreciate the acknowledgment of our algorithm’s efficiency and clear writing. We understand the reviewer's concerns regarding novelty and positioning. However, we respectfully claim that our main contribution lies in **the specific synthesis** of various components to solve a long-standing intractability in LGCPs, yielding theoretical guarantees (Global Concavity, Uniqueness) that are absent in the prior works cited.
>
> Below, we address the structural necessities of our design choices.
>
> **1. Novelty and Necessity of Voronoi Tessellation**
>
> Regarding the substantial advantages of using Voronoi tessellation over uniform grids or Monte Carlo integration, we emphasize that in our variational inference framework Voronoi tessellation is not only a quadrature choice, but a **structural requirement** for two reasons:
>
> *   **Alignment of Latent Variables with Prediction Targets:** A primary goal of spatial analysis is prediction at specific locations $\mathcal{S}$. Our Voronoi construction explicitly forces the integration nodes $\mathcal{I}$ to include the **prediction targets** $\mathcal{S}$ and **observed points** $\mathcal{D}$. This unifies the quadrature nodes with the inference targets. This allows us to infer the posterior exactly where needed without inflating the dimensionality of the latent field unnecessarily, for example, in uniform grid/lattice.
> *   **Deterministic Guarantees:** Unlike Monte Carlo integration, which introduces stochasticity into the ELBO, the Voronoi approach yields a **deterministic** objective function. It is the mathematical key that allows us to prove **Global Concavity** (Theorem 1) and **Uniqueness** (Theorem 2). These theoretical guarantees would be lost if we relied on stochastic MC or ad-hoc grid approximations.
>
> **2. Positioning: Spatial Covariates vs. Random Fields**
>
> In our proposed model (Eq. 1), we fully support and estimate spatial covariates effect $X(s)\beta$. However, our primary spatial learning focus is the latent Gaussian Random Field $Z(s)$ which captures the dependence that cannot be explained by spatial covariates.
>
> Existing works primarily on covariate estimation (e.g., [1,2,3] in the review) often treat the spatial dependence as a nuisance parameter. However, in scenarios like our "Hole Case" (Figure 3), the covariate information is insufficient. The model must rely on the latent field $Z(s)$ to bridge the gap. Our method is designed to be robust in these data-sparse regimes where the latent random field inference is critical, rather than just estimating $\beta$.
>
> **3. Positioning: Log-Link vs. Alternative Links (Square/Sigmoid)**
>
> The reviewer suggests comparisons to Gaussian Cox Processes with Square (Lloyd et al., 2015) or Sigmoidal (Donner & Opper, 2018) links.
>
> *   **Fundamental Difference:** We respectfully argue that these are structurally different models. The Log-Gaussian Cox Process (LGCP, Møller et al., 1998) is the canonical model because the log-link naturally ensures positivity ($\lambda > 0$) and offers a clear interpretation of effects.
> *   **Why others use Square/Sigmoid:** Methods utilizing Square or Sigmoid links were largely proposed to bypass the computational intractability of the log-link integral. All the referred literature [4,5,6] state that they change the model definition to make sparse GP inference easier. Therefore, the doubly intractable challenge for LGCP is long-standing and is not solved by changing different link functions.
> *   **Our Contribution:** We solve the inference problem for the **canonical LGCP** itself. We believe it is crucial to enable efficient fitting of the standard Log-link model rather than altering the link function solely for computational convenience. Therefore, a direct comparison to models with different link functions only assesses different modeling assumptions rather than inference quality.
>
> **4. Clarification on Low-Rank Approximation**
>
> While low-rank approximations are indeed standard, our contribution lies in a **hybrid strategy** that combines them with NNGP. Specifically, NNGP induces a **sparse precision matrix** $\Gamma$ to capture local neighboring dependence, while the Woodbury formula handles the global update via a low-rank perturbation. This results in an $O((N+n)^2 r)$ complexity. This hybrid structure is critical: it balances **statistical expressiveness** (capturing local variations that pure low-rank methods miss) with **computational feasibility**, effectively bypassing the $O((N+n)^3)$ bottleneck of full GP inference.

---

> > ### Comment · Reviewer_z2Fg · 2025-11-28
> >
> > Thank the authors for the detailed clarification. Below, I provide my comments on each of the authors’ main arguments.
> >
> > **Novelty and Necessity of Voronoi Tessellation: Alignment of Latent Variables with Prediction Targets**
> >
> > In the proposed Voronoi construction, the authors emphasize as an advantage that the integration nodes can always be chosen to include prediction and observation locations. However, it is unclear why this is important. In Gaussian process models, the intensity function can be estimated at arbitrary locations, regardless of where the integration nodes are placed. To the best of my knowledge, the locations of the integration nodes are tuned solely to improve the approximation accuracy of the integral term. If this understanding is incorrect, I would appreciate it if the authors could point out where I have made a mistake.
> >
> > **Positioning: Spatial Covariates vs. Random Fields**
> >
> > I now clearly understand the intended positioning of the present work. However, I think there is one misunderstanding on the authors’ side.
> >
> > The authors argue that in existing works (e.g., [1,2,3]), spatial dependence is often treated as a nuisance parameter, but this is an incorrect way of understanding. If we recall that one can include the spatial location itself as an element of the covariate vector, i.e., $X_i(s) = s$, then it becomes clear that these existing studies are in fact generalizations of latent Gaussian random field models on the observation space. In other words, they are not treating the spatial variable as a non-essential auxiliary parameter; rather, they are treating the spatial variable on an equal footing with other covariates.
> >
> > By contrast, in the proposed approach, covariates $X(s)$ are explicitly distinguished from the spatial variable by imposing log-linear dependence on the former. This is (presumably) rooted in the tradition of Cox proportional hazards models, which is a well-established and respectable line of work; there are various pros/cons to this choice, which I will not go into here. Depending on the application, either approach may be more appropriate. For the “hole case” at least, I am not convinced that one can already say which of the two paradigms is preferable.
> >
> > **Positioning: Log Link vs. Alternative Links (Square/Sigmoid)**
> >
> > Even after this explanation, I am still unsure why LGCPs should be given a special status. In particular, the authors emphasize that the effects are easier to interpret under the exponential link function, but I do not see why interpretability should be impaired under a sigmoidal link function: both are nonlinear, monotonically increasing functions. While it is true that the exponential link function has a long history, I am not aware of reasons beyond this historical aspect that would justify treating LGCPs as fundamentally more important than models with alternative link functions. The authors should clarify this point more explicitly, or alternatively, provide a comparative discussion with recent efficient Gaussian Cox process models that employ other link functions.
> >
> > For the reasons above, I will maintain my score.

---

> > > ### Author Response · Authors · 2025-12-01
> > >
> > > We thank the reviewer for the follow-up. We appreciate the opportunity to clarify these thoughtful concerns.
> > >
> > > **1. Novelty & Necessity of Voronoi (Alignment of Nodes)**
> > >
> > > The reviewer is theoretically correct: a GP defined on a continuous domain can indeed be predicted at any location $s^*$ via Kriging, regardless of the integration mesh.
> > > However, our motivation for aligning integration nodes with prediction targets is computational efficiency under the framework of variational inference and NNGP.
> > >
> > > *   **The Computational Bottleneck:** In our NNGP-based inference, the precision matrix $\Gamma$ is sparse and defined over the set of nodes $\mathcal{I} \cup \mathcal{D}$. If we wish to predict at a new set of locations $\mathcal{S}$ apart from this set, additional interpolation step or Kriging is needed. This extra step may introduce further approximation error and thus potentially weaken the computational efficiency and theoretical guarantees.
> > > *   **Our Solution:** By constructing the Voronoi mesh such that $\mathcal{S} \subset \mathcal{I}$, the prediction targets become native nodes in the sparse NNGP graph. This allows us to obtain posterior predictions directly from the variational optimization output without additional Kriging step. This unifies training and prediction into a single and efficient framework.
> > >
> > > **2. Positioning: Spatial Covariates vs. Random Fields**
> > >
> > > Our use of *nuisance parameter* is misleading. We clarify that our framework explicitly adopts a **mixed-effect model** perspective, which distinguishes between these components for specific statistical reasons:
> > >
> > > *   **Fixed vs. Random Effects:** We treat the spatial covariates $X(s)\beta$ as **fixed effects** (modeling deterministic, global mean trends) and the Gaussian Process $Z(s)$ as **spatial random effects** (modeling local, stochastic correlation). In particular, the spatial covariates can be flexibly constructed using domain knowledge, such as nonlinear functions of $s$, geological information, and realizations of other spatial processes. It is more like feature engineering in regression design.
> > > *   **Insufficiency of Parametric Trends:** While including location as a covariate (e.g., $X(s)=s$ or polynomial trends) can model global pattern, it acts as a purely parametric trend surface. This is often **insufficient** for modeling complex spatial point patterns, which frequently exhibit local clustering driven by unobserved factors that a global function of coordinates cannot easily capture.
> > > *   **Advantage in the "Hole Case":** Methods that model intensity as a deterministic function often struggle to quantify uncertainty in large unobserved gaps (holes), sometimes extrapolating trends too confidently. In contrast, our mixed-effect approach allows the random effect $Z(s)$ to revert to the prior indicating high uncertainty, while the fixed effect $X(s)\beta$ provides a stable baseline. This conservative behavior in data-sparse regions is consistent with the intuition of uncertainty quantification.
> > >
> > > **3. Positioning: Log Link vs. Alternative Links (Square/Sigmoid)**
> > >
> > > As our paper’s title suggests explicitly, our specific goal is to provide an efficient variational method for the canonical point pattern model **LGCPs** (Møller et al., 1998). We successfully resolve the long-standing computational challenges associated with fitting the LGCP itself, rather than exploring alternative link functions.
> > >
> > > *   **Advantages of the Log-Link:** We prioritize the Log-link because it offers specific modeling advantages that other links lack:
> > >     1.  **Multiplicative Interpretation:** In domain sciences such as epidemiology and ecology, the log-link provides a **Relative Risk** interpretation. For example, covariate $X$ increases intensity by a factor of $e^\beta$. This multiplicative structure is lost with Square or Sigmoid links.
> > >     2.  **Unbounded Support:** The log-link maps $\mathbb{R} \to [0, \infty)$. In contrast, Sigmoidal links map to a bounded range $[0, \lambda_{max}]$, imposing an artificial saturation ceiling that is physically inappropriate for many phenomena.
> > > *   **Methodological Contribution:** Alternative links were introduced in the machine learning literature largely to **bypass** the analytical intractability of the log-intensity integral. While they make inference easier, they fundamentally alter the generative model. Our work accepts the challenge of the log-link integral directly and solves it, rendering the need to switch to these "computational convenience" models unnecessary.

---

### Official Review · Reviewer_wqsK · 2025-11-01

**Soundness:** 3
**Presentation:** 4
**Contribution:** 3
**Rating:** 6
**Confidence:** 2

**Summary:**

This paper proposes an efficient variational inference method for Log-Gaussian Cox Processes (LGCPs). Instead of relying on expensive MCMC, the authors use a Voronoi-based Poisson likelihood approximation, a Gaussian variational family, and a coordinate-ascent optimization scheme. They provide theoretical guarantees, including concavity, existence/uniqueness of the optimum, and convergence. Further computational speedups are achieved using nearest-neighbor GP approximations and matrix identities. Experiments on synthetic and real datasets show comparable or superior accuracy to existing LGCP methods with significantly improved efficiency and robustness to missing spatial regions.

**Strengths:**

* The paper contributed to scalable Bayesian spatial modeling, with a novel combination of Voronoi approximation + VI + NNGP for LGCP inference.
* The paper has solid theoretical foundations with clear convergence and optimality guarantees.
* The paper provided a practical algorithm that is computationally efficient and implementable.

**Weaknesses:**

1. Some theoretical sections are dense, and intuition can be made more straightforward.
2. Limited discussion of performance on very large-scale datasets or high dimensions.
3. Comparisons focus mostly on classical spatial models, but deep generative point-process baselines are absent.

**Questions:**

* How scalable is the approach to very large spatial domains or high-resolution grids?
* Can the method be extended to spatiotemporal LGCPs or non-stationary kernels?
* How sensitive is performance to choices in Voronoi partitioning and variational family?

---

> ### Author Response · Authors · 2025-11-22
>
> We thank the reviewer for the encouraging assessment, particularly for highlighting the novel combination of our methods and our solid theoretical foundations. We are glad the reviewer found the presentation excellent. We address the specific questions regarding baselines, scalability, and extensions below.
>
> **1. Comparison to Deep Generative Baselines**
>
> The reviewer noted the absence of deep generative point process baselines. We focused our comparison on INLA and VIFRK because our goal is to solve the **specific inference problem of LGCPs**, which are favored in scientific domains (ecology, epidemiology) for two reasons that deep models often lack:
>
> 1.  **Interpretability:** The separation of the linear predictor $X(s)\beta$ and the random field $Z(s)$ allows researchers to explicitly quantify the effect of covariates versus spatial correlation.
> 2.  **Uncertainty Quantification (UQ):** LGCPs provide a rigorous probabilistic posterior map. Deep generative models for point processes often focus on density estimation but struggle to provide calibrated credible intervals for the intensity surface in spatial domains with missing data, e.g., our "Hole case" numerical experiment.
>     However, we agree this context is important. We will add a discussion in the "Related Work" section clarifying why variational LGCPs are preferred over black-box deep generative models when UQ and data-efficiency are priorities.
>
> **2. Scalability to Challenging Domains (Question 1)**
>
> The scalability of our approach is determined by the number of mesh nodes ($n$) and observation points ($N$).
>
> *   **Computational Cost:** Thanks to the NNGP approximation and the Woodbury identity (Eq. 18), the dominant cost per iteration is $O((n+N)^2r)$, where $r$ is the number of neighbors (usually small). It is a significant saving in computation, rather than the cubic $O((n+N)^3)$ of standard GPs.
> *   **High Resolution:** We can handle high-resolution grids easily. In our Real Data experiments (Table 2), we successfully modeled transcriptomics data with thousands of cells. The memory footprint is sparse, allowing scaling to $N \sim 10^5$ or $10^6$ on standard hardware, significantly larger than what standard MCMC or exact GP methods can handle.
>
> **3. Extensions to Spatio-temporal / Non-stationary / High-dimensional Models (Question 2)**
>
> *   **Spatio-temporal:** Yes, the method extends naturally. A spatio-temporal LGCP is simply a GP over $\mathbb{R}^3$ (space + time). The Voronoi tessellation would become a 3D tessellation, and the NNGP neighbor search would occur in space-time. The optimization algorithms (Newton + Fixed Point) would remain mathematically identical.
> *   **Non-stationarity:** The core algorithm is agnostic to the kernel function $K_\theta$. We used a stationary Matérn kernel for standard benchmarking, but one could substitute a non-stationary covariance function into the NNGP construction (Eq. 31) without changing the variational update rules.
> *   **High Dimensions:** We distinguish between two types of dimensionality:
>     *   **High-Dimensional Covariates ($\mathbf{X}$):** Our method handles high-dimensional feature spaces very efficiently. The update for the regression coefficients $\beta$ (Eq. 13) relies on Newton’s method, which scales robustly with the number of covariates. The high-dimensional setting in sparse learning where the covariate dimension is larger than the sample size is unrealistic in spatial modeling and is thus out of the scope of current work.
>     *   **High-Dimensional Domain ($s$):** As mentioned above, our method adapts to spatio-temporal models of dimension 3 or 4. We note that for dimensions significantly higher, the Voronoi tessellation becomes computationally infeasible due to the *Curse of Dimensionality*. However, LGCPs are primarily employed in **physical spatial statistics** (geostatistics, ecology, epidemiology), which are inherently restricted to $\mathbb{R}^2$ or $\mathbb{R}^3$, this geometric limitation does not hinder the method's applicability to its core target problems.
>
> **4. Sensitivity to Voronoi Partitioning (Question 3)**
> As noted in our response to Reviewer 2, the Voronoi partition acts as a numerical quadrature rule. The sensitivity follows standard finite element analysis results (Simpson et al., 2016): as long as the mesh density is sufficient to resolve the length-scale of the Gaussian Process, the approximation error is negligible compared to the statistical uncertainty. We use standard "Dual-Mesh" construction strategies to ensure this robustness.
>
> **5. Improving Intuition of Theoretical Sections**
> We appreciate the feedback that the theory is dense. In the final version, we will add "Proof Sketches" at the beginning of the Appendix to provide high-level intuition before diving into the rigorous algebra.

---

### Official Review · Reviewer_ZLDd · 2025-11-01

**Soundness:** 3
**Presentation:** 3
**Contribution:** 3
**Rating:** 4
**Confidence:** 4

**Summary:**

The authors focus on modeling spatial point patterns using a Log-Gaussian Cox Process (LGCP). In this model, the point pattern $Y$ follows an inhomogeneous Poisson process, conditional on an intensity surface $\lambda(s)$. The logarithm of this intensity surface is modelled as the sum of a linear predictor and a latent Gaussian random field:

$$log \lambda (s) = X(s)\beta + Z(s); Z(s)\sim GP(0, K_{\theta}(s,s'))$$

This doubly stochastic, hierarchical construction provides the LGCP with greater flexibility than standard inhomogeneous Poisson models.
- The primary challenge lies in fitting the LGCP and inferring the log-intensity surface, $\lambda(s)$, which models the expected number of points per unit area.
- A typical approach to this problem is using Markov Chain Monte Carlo (MCMC) methods. However, these methods are often burdened by substantial computational costs and difficulties in assessing convergence, limiting their scalability. Other leading approaches include Integrated Nested Laplace Approximation (INLA) and variational inference methods.The authors of this paper have chosen to develop a variational inference framework to better fit the LGCP model.

The key ideas used in the paper are:
1. authors pre-select n auxiliary integration points within the observation area on the top of the given N observations; using Delaunay triangulation, the authors construct Voronoi tesselation; authors hypothesise that this leads to more accurate and efficient inference of LGCP.

2. The artificial extension creates computational burden as ELBO evaluation is carried on (n+N) points; authors propose to speed up the computation by approximating the Gaussian field in the intensity model via Nearest-Neighbour Gaussian Process, there the full conditional distribution is replaced by the the distribution conditioned only on the set of the nearest neighbours.

3. Authors discuss in detail how to apply Newton Optimisation and how to update the variational covariance estimate in detail. For the covariance estimation, authors propose using Woodbury formula. It is not clear to me whether this is innovative in connection with Gaussian processes estimation.

Briefly, the method works as follows:
 - the logprob to estimate the model $p(Y| \lambda)$ contains the product of the intensity values at each observation point and a multiplier corresponding to Poisson term where the intensity is an integrated over the entire domain $\Omega$.
$$p(Y|\lambda) = exp\Big( - \int_{\Omega} \lambda(s) ds \Big) \Pi_{i=1}^N \lambda(s_i)$$
The integral term over the domain $\Omega$ is evaluated on the grid of Voronoi tessellation; the product term corresponding to likelihood of the intensity itself is evaluated on the observed points
-  the prior Gaussian Process is approximated by NNGP
- the linear covariate parameter $\beta$ and the mean /drift  $\mu$ of the Gaussian field is computed uby the Newton method
- the covariance matrix of the variational approximation is computed using the matrix inversion
- the method also optimises the likelihood noise parameter

**Strengths:**

1. Paper is reasonably well written with good logical structure and easy to follow.
2. The ideas seems to be novel and in general interesting.
3. Authors provide optimisation algorithm and attempt to safe-guard the proposed methods by theoretical results.
4. Well described optimisation algorithm for fitting the parameters.

**Weaknesses:**

1. The main contribution of the paper is, in my opinion, in selecting the auxiliary points to improve the integration within the likelihood. While this idea is interesting, authors do not elaborate on how to select these auxiliary points and does not provide any ablation study on this topic. Given that this the major step, I consider this as a crucial weakness. It would be great to see experiment, where the the authors demonstrate the robustness / sensitivity of the method to these auxilary points.
2. Authors claim they prove theoretical properties of the estimator. Proof for Theorem 1 is only sketched, while for Theorem 2, the provided proof only shows existence of the solution but does not prove the uniqueness.
3. Paper does not provide experiments showing how the NNGP selection impacts the inference method (in comparison to using the GP without any approximation on n+N datapoints).
4. The experimental results does not seem to over-perform the existing methods. It is not major weakness as authors provide interesting algorithm but I believe that benefits of the proposed should be somehow better demonstrated than by results in Figure 3., e.g. authors would consider to construct a metric that can demonstrate how to measure depicted phenomena.

Minor typos:
l 116: dimension m is not defined
l 151: dimension p is not defined in Z_s  / defined on the l 182, slightly confusing

**Questions:**

1. How do you select the “pre-selected” deterministic integration points (l 108)? How sensitive are the results to this selection? Can you provide an ablation study to demonstrate this?

2. When fitting a Gaussian process, the combination of the hyperparameters is not unique, only the eigen-values of the covariance matrix. Can you elaborate on how this impacts your fits?

3. When approximating GP with NNGP (l 201), how does this impact the results from the Theorems 1 and 2? What is the impact of NNGP approximation on the performace on the method?

4. For Newton method, how do you select the initial values (l 242)?

5. How numerically stable are the inversions (18) - l 281

---

> ### Author Response · Authors · 2025-11-22
>
> We thank the reviewer for the detailed and accurate summary of our work. We appreciate the recognition of our novelty. Below, we address the concerns regarding the auxiliary integration points, theoretical proofs, and experimental comparisons.
>
> **1. Selection and Sensitivity of Auxiliary Integration Points**
>
> We agree that the auxiliary integration points are crucial.
>
> *   **Selection Strategy:** As described in Section 2.1 (lines 133-138), we do not select these points arbitrarily. We utilize the **Dual-Mesh** strategy established in the INLA-SPDE literature (Simpson et al., 2016) and ensure prediction locations are among the vertices. Specifically, the integration points correspond to the vertices of a Delaunay triangulation generated over the domain. The weights $w_i$ are the areas of the Voronoi tiles dual to these vertices.
> *   **Sensitivity:** The accuracy of the integral approximation $\int \lambda(s) ds \approx \sum w_i \exp(Z(s_i))$ is determined by the mesh density (triangle size). Simpson et al. (2016) theoretically proved that as the mesh size approaches zero, the approximate posterior converges to the true posterior in Hellinger distance.
> *   **Ablation:** In our experiments, we set the integration points to coincide with the prediction locations (dense grids) to ensure high accuracy. We are happy to include an ablation study in the appendix of the final version, but existing theory suggests that as long as the mesh is sufficiently fine to capture the spatial correlation scale, the results are robust.
>
> **2. Clarification on Theoretical Proof on Uniqueness**
>
> We respectfully point out that **uniqueness is a direct consequence of Theorem 1**.
>
> Theorem 1 proves that the objective function $E(\beta, \mu, \Sigma)$ is **strictly jointly concave**. Theorem 2 proves the function is **coercive** (approaches $-\infty$ at boundaries). It then follows from standard convex analysis that the optimizer exists and must be unique. We will make this statement more explicit in Appendix B.
>
> **3. Impact of NNGP on Theory and Performance**
>
> *   **Impact on Theorems:** The approximation of the full GP with an NNGP replaces the dense precision matrix with a sparse precision matrix $\Gamma$ which is still **positive definite**. Therefore, Theorems 1, 2, and 4 (concavity and convergence of the algorithm) hold exactly the same whether we use a full GP or NNGP. The optimization landscape remains strictly concave.
> *   **Impact on Performance:** The NNGP introduces a modeling bias or approximation error in exchange for improved computational scalability. However, Datta et al. (2016) showed that NNGP serves as an excellent approximation to full GPs for spatial data compared with existing low-rank and/or sparse approximation alternatives (Heaton et al., 2019). Our choice of NNGP is practical: Full GP inference is computationally infeasible for the large datasets we target, making NNGP necessary.
>
> **4. Experimental Performance and Metrics**
>
> We respectfully refer the reviewer to **Table 1**, which reports the **Predictive Log-Likelihood (pred loglik)**. This is the standard proper scoring rule for probabilistic predictions.
>
> In the "Hole case", our method achieves a predictive log-likelihood of **1958.95**, significantly outperforming INLA (**749.41**) and VIFRK (**904.28**). This quantitative metric confirms the visual improvement seen in Figure 3.
>
> In **Table 2**, our method outperforms under the predictive log-likelihood metric as well for two real-world data.
>
>
> **Answers to Minor Questions**
>
> *   **Initial Values (Line 242):** We initialize $\mu$ and $\beta$ as constant one vector and $\Sigma$ as a constant diagonal matrix. Due to the global concavity (Theorem 1), the algorithm converges to the unique optimum regardless of initialization.
> *   **Stability of Inversions (Eq 18):** The matrix being inverted is $(D_k^{-1} + \sigma^{-2} L^T D_k^{-1} L)$. Since $D_k$ is diagonal and positive, and $L$ comes from the Cholesky of the precision matrix, this core matrix is symmetric positive definite and well-conditioned, ensuring high numerical stability compared to direct covariance inversion.
> *   **Hyperparameter Non-uniqueness:** We resolve the identifiability issue by fixing the range/smoothness parameters using empirical variograms, a common practice in spatial statistics, and only optimizing the marginal variance $\sigma^2$ within the variational loop.
> *   Thanks for pointing out the minor typos.

---

### Official Review · Reviewer_jtYa · 2025-11-01

**Soundness:** 3
**Presentation:** 3
**Contribution:** 2
**Rating:** 6
**Confidence:** 3

**Summary:**

This paper proposes and studies an variational approach to fit log-Gaussian processes.

**Strengths:**

The paper is well-written, and I mostly enjoyed reading it. The idea is clear, and the results are scientifically correct. The numerical experiments also support the proposed approach.

**Weaknesses:**

The paper is more a statistical paper, rather than a ML/AI work. Though the results seems to be novel, the approaches (and the proofs) are standard.

**Questions:**

See weaknesses.

---

> ### Author Response · Authors · 2025-11-21
>
> Thanks for the positive assessment of our paper’s clarity, scientific correctness, and the strength of our numerical experiments. We appreciate the constructive feedback regarding the positioning of our work. Our work falls within the primary area scope of **probabilistic methods**, given its focus on Bayesian analysis, variational inference, and uncertainty quantification.
>
> Below, the concerns regarding the paper’s fit for the ML community and the novelty of our approach are addressed.
>
> **1. On the fit for ICLR (Statistics vs. Machine Learning)**
> While Log-Gaussian Cox Processes (LGCPs) originate in spatial statistics, they are fundamentally **generative models for point pattern data**, a core subject of interest in machine learning. We respectfully argue that this work fits within ICLR’s scope for two reasons:
>
> *   The primary bottleneck in modern Gaussian Process research is scalability. Our work transforms a computationally heavy inference problem (traditionally solved via MCMC) into a highly efficient optimization problem via variational Inference. Under the Bayesian framework, this focus on computational efficiency, coordinate ascent algorithm, and theoretical justification aligns with the core interests of the community.
> *   There is a strong tradition of Point Process research at top ML venues (e.g., Lloyd et al., *ICML 2015*; Aglietti et al., NeurIPS 2019). Our work advances this lineage by solving the long-standing challenges in existing literature using variational inference and NNGP.
>
> **2. On the novelty of the approach and proofs**
> The reviewer noted that the approaches appear standard. We wish to clarify that while the components (Voronoi tessellation, variational inference, Newton’s method) are established, their specific integration and the resulting theoretical guarantees are novel and non-trivial:
>
> *   **Likelihood Formulation and Global Concavity:** Proving the existence and uniqueness of the optimal solution (Theorems 1 and 2) for the LGCP ELBO is not guaranteed in general settings. Our derivation utilizing the Voronoi tessellation provides a theoretical justification that other "black-box" variational inference methods lack.
> *   **Coordinate Ascent Maximization Algorithm:** We do not simply apply off-the-shelf optimization. Considering the specific nature of parameter blocks, we carefully designed optimization strategies. Notably, we derived a **novel fixed-point updating rule** for the variational covariance. We proved this operator is anti-monotonic and contractive (Theorem 4 and Lemma 3), ensuring convergence. This is a novel algorithmic contribution, not a standard application.

---

### Meta-Review · Area_Chair_7QVp · 2026-01-12

**Summary:**

The paper as is has several issues, which were brought up by reviewers, and with which I agree.

*Insufficient discussion of relation to existing work*: Many approaches to inference in LGCP exist, as discussed in lines 50-72. However, the authors fall short in explicitly discussing the problems with these existing methods. They simply continue to state their approach, without justifying _why_ the approach they choose is better than any of the existing approaches. Each existing approach already addresses an element of what this paper addresses (e.g. scalability of the GP, intractability of the likelihood etc).

As such, the main justification for accepting the paper lies in the improved log likelihood performance. However, if one is just interested in improving log likelihoods, then why dismiss other point processes, e.g. that use different link functions? I agree with the reviewer z2Fg, that from this perspective, the LGCP should not be given special status, and other link functions (e.g. sigmoid, or square, as John & Hensman did) should be benchmarked against. The fact that the model is "structurally different" does not matter when the goal is simply to get good predictive log-likelihoods.

In the discussion phase, the authors mentioned two justifications for their approach that were not clear in the paper.

Firstly, the deterministic nature of their approach. They mention their two theorems about this. This is all good and well, but why are these theorems relevant to solving the problem of LGCPs? The authors have not justified this in the paper or the discussion. It's nice to have guarantees, but these guarantees need to be relevant to solving the problem. Stochastic approximations are also guaranteed to converge (Robbins-Munro), so why are the guarantees provided here better?

Secondly, the authors mention a desire to assess inference quality, rather than modelling assumptions. This is an excellent point, and a worthy cause for a paper. However, if this was the goal of the paper, then the experiments still fall short, since the only evidence that is provided that inference is better is better log likelihoods. The only experiment that gives _some_ evidence for this is the first synthetic experiment. Since the ground truth was sampled from the model, on average over many sampled datasets the true posterior should give the best predictive log likelihoods. However, the quality of the posterior is not directly assessed for the real datasets.

To summarise: Reviewers gave many good points, and to improve this paper, I would recommend:
- comparing to more point process methods, including ones with other link functions, if the argument is that the method is good because it gives better log likelihoods
- perform more in-depth comparisons on the quality of inference, if the argument is that the method is interesting because it gives a better posterior approximation in the LGCP specific model (perhaps by comparing to asymptotically-exact MCMC methods like the Murray paper, or alternatively the Gaussian Process Density Sampler, which both address the doubly stochastic issue).

**Reviewer Concerns:**

See above.

**Reviewer Scores:**

Some issues were still remaining. The most strongly opinionated reviewer engaged with the rebuttal, and so is unlikely to have changed much given more time.

---

### Decision · Program_Chairs · 2026-01-26

Reject